# FULLY QUANVOLUTIONAL NETWORKS FOR TIME SERIES CLASSIFICATION

## ABSTRACT

Quanvolutional neural networks have shown promise in areas such as computer vision and time series analysis. However, their applicability to multi-dimensional and diverse data types remains underexplored. Existing quanvolutional networks heavily rely on classical layers, with minimal quantum involvement, due to inherent limitations in current quanvolution algorithms. In this study, we introduce a new quanvolution algorithm that addresses previous shortcomings related to performance, scalability, and data encoding inefficiencies. Specifically targeting time series data, we propose the Quanv1D layer, which is trainable, capable of handling variable kernel sizes, and can generate a customizable number of feature maps. Unlike previous implementations, Quanv1D can seamlessly integrate at any position within a neural network, effectively processing time series of arbitrary dimensions. Our chosen ansatz and the overall design of Quanv1D contribute to its significant parameter efficiency and inherent regularization properties. In addition to this new layer, we present a new architecture called Fully Quanvolutional Networks (FQN), composed entirely of Quanv1D layers. We tested this lightweight model on 20 UEA and UCR time series classification datasets and compared it against both quantum and classical models, including the current state-of-the-art, ModernTCN. On most datasets, FQN achieved accuracy comparable to the baseline models and even outperformed them on some, all while using a fraction of the parameters.

## 1 INTRODUCTION

As Moore's Law approaches its limits, a global shift is underway toward quantum computing–an alternative capable of solving challenges intractable for classical systems (Nielsen & Chuang, 2010). Given the practical implications of machine learning and the widely sought-after "quantum advantage" offered by quantum computing and its algorithms, quantum machine learning (QML) is gaining traction quickly (Biamonte et al., 2017). For instance, the domain has already seen the implementation of several quantum adaptations of support vector machines (Rebentrost et al., 2014; Li et al., 2015) and neural networks (Tacchino et al., 2019; 2020). In this paper, we revisit one such QML algorithm, the quanvolution algorithm (Henderson et al., 2020), analyze its utilities and pitfalls, and propose an improved version that can overcome performance, scalability, and modularity issues in existing networks.

### 1.1 BACKGROUND

In the literature, there are two variants of quantum convolution: circuit-based and kernel-based (also referred to as quanvolution). With different quantum operations, the circuit-based variant mimics the Conv2D and pooling layers to turn a convolutional neural network (CNN) into a quantum circuit (Cong et al., 2019; Hur et al., 2022). However, unlike a typical convolution operation, this variant works with the entire flattened input instead of individual input patches. Although the entire process can be executed on a quantum computer, the scalability of circuit-based quantum convolution compared to classical CNNs remains open for debate. For instance, this variant struggles to encode large, complex image datasets in this noisy intermediate-scale quantum (NISQ) era. Additionally, the reliance of neural networks on nonlinearities conflicts with quantum mechanics.

In contrast, kernel-based quantum convolution, also known as quanvolution, functions similarly to a single classical convolutional layer (Henderson et al., 2020). Essentially, the quanvolution operation is the result of quantum circuits substituting for typical filters or kernels inside a convolutional layer. This approach offers greater flexibility compared to the circuit-based variant, enabling the development of hybrid models and training schemes. In a classical-quantum hybrid model, a quantum computer deals with the primary computations–interactions between input patches and filters–while classical computers handle tasks like processing loss values and applying nonlinear activations. Owing to its versatility, the algorithm's introduction remains a staple in quantum-based computer vision with wide range of applications (Ullah & Garcia-Zapirain, 2024; Kharsa et al., 2023; Zhang et al., 2024; Yang et al., 2021; Peral-García et al., 2024).

This study focuses on the kernel-based quantum convolution, or quanvolution, and aims to address various limitations outlined in the following section.

## 1.2 MOTIVATION

Quanvolutional neural networks are implemented as hybrid systems, where learning is achieved through the combined effort of quantum and classical layers. In such architectures, the initial convolution layer in models like LeNet-5 (LeCun et al., 1998) is replaced with a quanvolutional layer, while the rest of the network remains unchanged. However, most implementations heavily rely on deep classical layers, typically including only a single quantum layer. This raises questions about the true contribution of quanvolutional layers–whether they play a meaningful role or if the classical layers shoulder the heavy lifting. As such, quanvolutional neural networks are very limited in both applicability and performance, and these limitations stem from the quanvolution algorithm itself.

The original quanvolution algorithm was initially designed to work only with single-channel image patches (Henderson et al., 2020). While some studies have managed to extend its application to RGB or three-channel images (Jing et al., 2022; Savla et al., 2022), the algorithm still lacks the capability to process 2D data with an arbitrary number of channels, unlike a classical Conv2D layer. In modern deep learning architectures, the number of channels or feature maps typically increases as the network deepens. For instance, EfficientNet-B0 starts with a three-channel input image and expands to a feature dimension of 1280 in the final convolutional layer (Tan & Le, 2019). Achieving this level of scalability remains a challenge for existing quanvolution algorithms.

Moreover, a quanvolutional layer produces only a limited number of feature maps, primarily due to its small kernel size and reliance on a single filter. Generally, the kernel size influences the number of wires in the circuit, which in turn affects the number of feature maps. Such an approach is in stark contrast to classical convolution. For Conv2D layers, users can adjust kernel sizes to capture either fine-grained details or broader contextual features as needed. Furthermore, classical layers utilize multiple kernels or filters to extract varied but pertinent patterns from the same input, improving the model's ability to generalize. As such, when compared to classical convolution, the quanvolution algorithm has yet to reach its overall performance potential due to the rigidity in kernel size and circuit selection.

Our motivation for this work was to create a learnable quanvolutional layer that works like modern convolutional layers and gives users the freedom to choose the kernel size and number of output feature maps. Our aim was to ensure that this layer could be seamlessly integrated into any network, provided the input dimensions were compatible, mirroring the versatility of conventional convolution. However, we recognized that making a practical, scalable, and modular 2D quanvolutional layer right away would be very challenging because of NISQ-related limitations. As a result, we opted to begin with its 1D counterpart as a foundational step.

## 1.3 CONTRIBUTIONS

As a main contribution, this study introduces the Quanv1D layer, a quantum analog to the Conv1D layer. Similar to Conv1D, Quanv1D supports multichannel data, variable kernel lengths, and adaptable feature map generations. While designing this layer, we prioritized efficiency and practicality, taking into account the limitations of current quantum hardware and aiming to minimize computational costs. For instance, based on the input requirements and desired output, the layer adjusts itself by either using a higher number of qubits with fewer circuits or reducing the number of qubits

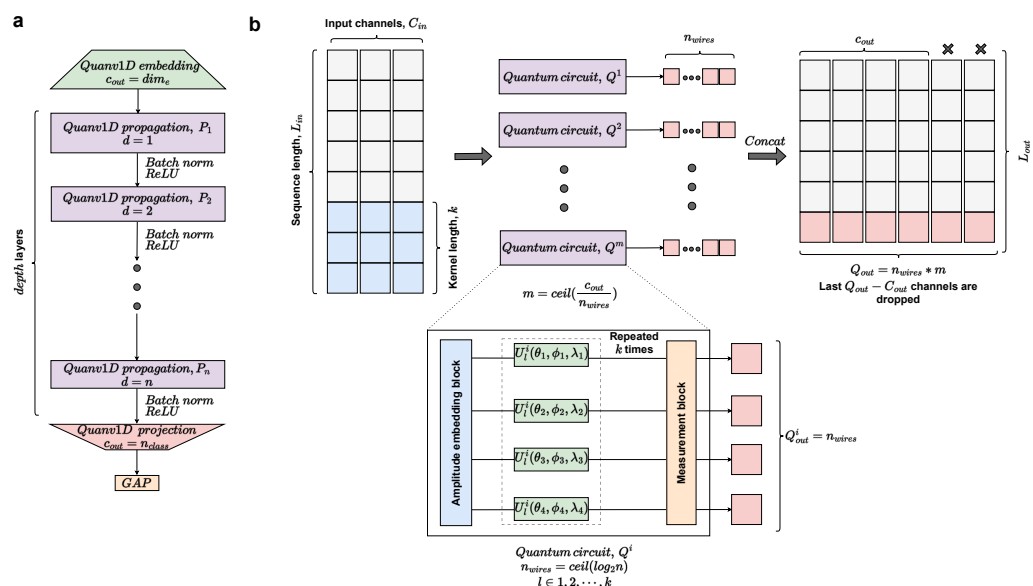

Figure 1: (a) FQN's overall architecture. It has three main parts–embedding, propagation, and projection–consisting of Quanv1D layers. For learning stability and non-linearity, we included batch normalization and ReLU activation in the architecture. GAP denotes global average pooling. (b) Workflow inside the Quanv1D layer. Unlike a conventional filter found in a convolutional layer, Quanv1D uses quantum circuits to generate feature maps.

while increasing the number of filters. Additionally, we reduced the qubit usage for scalable data encoding, which resulted in a single filter requiring $log_2(C_{in} * k)$ qubits instead of $C_{in} * k$ qubits stemming from linear mapping methods. Here, $C_{in}$ represents the input channel size and $k$ represents the kernel length. Furthermore, thanks to our carefully chosen ansatz, Quanv1D exhibits a self-regularizing property that enhances training performance.

Our secondary contribution lies in designing an efficient quanvolutional neural network for time series classification. As of now, quanvolution in time series analysis remains underexplored. Most existing approaches adapt image-based quanvolution by first converting time series data into visual representations, such as scalograms or spectrograms (Savla et al., 2022; Li et al., 2024; Sridevi et al., 2022; Yang et al., 2021; Prabhu et al., 2023). Also, the only direct 1D quanvolutional approach, proposed by Rivera-Ruiz et al. (2023), was limited to univariate time series data. All these methods suffer from over-reliance on classical layers, scalability, limited feature maps, and rigid kernel size, as discussed earlier. To overcome these limitations, we present a fully quanvolutional network (FQN) built only with our Quanv1D layer. This design not only resolves the identified challenges but also demonstrates the potential of stacked quantum layers to enhance representation learning in temporal data. Moreover, FQN is very lightweight, requiring substantially fewer trainable parameters than its quantum and classical counterparts.

We provide detailed technical explanations of Quanv1D and FQN's design specifics in the following section. In Appendix A, we discuss some quantum computing basics, alongside how our proposed 1D quanvolution differs from the existing one.

## 2 METHOD

### 2.1 QUANV1D

In general, Conv1D accepts input in the form $(N, C_{in}, L_{in})$, where $N$ represents the batch size, $C_{in}$ is the number of input channels or dimensions, and $L_{in}$ refers to the sequence length. The output shape is $(N, C_{out}, L_{out})$, where $C_{out}$ is a user-defined parameter specifying the number of output

channels following the quanvolution operation. The value of $L_{out}$ is computed using the following equation:

$$L_{out} = \left\lfloor \frac{L_{in} + 2 \times p - d \times (k-1) - 1}{s} + 1 \right\rfloor \tag{1}$$

Here, $k$ stands for the kernel size, $s$ for the stride, $p$ for the zero padding on both sides of the input, and $d$ for the spacing between kernel points. Quanv1D has been designed such that it mimics a Conv1D layer and follows the same patching operations (Chellapilla et al., 2006). For this reason, we use the hyperparameters presented in Equation (1) to determine the patches that will serve as input to the quanvolutional filters. These hyperparameters, such as the kernel length, also affect quantum-related calculations inside the layer, like how many quantum filters there are, how many qubits are in a single filter, how many unitary operations are in an ansatz, and so on.

In this study, amplitude embedding is used to convert the classical information from the extracted patches into a quantum feature space (Schuld, 2018). This method encodes $2^n$ features into the amplitude vector of $n$ qubits, as shown in the following equation:

$$|\psi\rangle = \sum_{i=1}^{2^n} \alpha_i |i\rangle \tag{2}$$

In this equation, $\alpha_i$ are the elements of the amplitude vector $\alpha$, and $|i\rangle$ represent the computational basis states. Each quanvolutional filter takes an input of $C_{in} \times k$ features, requiring $n = \lceil \log_2(C_{in} \times k) \rceil$ qubits for encoding. Before encoding, the features are normalized using $\sqrt{softmax(\alpha)}$ to ensure $|\alpha|^2 = 1$. Additionally, if $C_{in} \times k$ is smaller than $2^n$, the features are padded with zeros after normalization to match the required dimension size. We chose amplitude embedding over the usual linear mapping method, such as angle encoding, to reduce qubit usage and achieve compact data representation. This ensures the modularity we tried to achieve, which, otherwise, would be difficult to achieve with angle encoding, as it demands an impractically large number of qubits, even for simulations.

The matrix form of the chosen unitary operator, $U$, is given by:

$$U(\theta, \phi, \lambda) = \begin{pmatrix} \cos(\theta\frac{\pi}{2}) & -e^{i\lambda}\sin(\theta\frac{\pi}{2}) \\ e^{i\phi}\sin(\theta\frac{\pi}{2}) & e^{i(\phi+\lambda)}\cos(\theta\frac{\pi}{2}) \end{pmatrix} \tag{3}$$

Here, $\theta$ and $\lambda$ are trainable, while $\phi$ is fixed but initialized randomly. This is because, although $\phi$ is essential for introducing phase shifts within the circuit, its gradient during parameter updates is theoretically derived to be zero (refer to Appendix F). Consequently, it remains static throughout the optimization process.

For an $n$-qubit circuit, the unitary operator is applied to each qubit, forming a layer of unitaries, and this layer is repeated $k$ times. The total unitary operations can be expressed as follows:

$$U_{\text{total}} = \prod_{l=1}^{k} \left( \bigotimes_{i=1}^{n} U_{il}(\theta_{il}, \phi_{il}, \lambda_{il}) \right) \tag{4}$$

Here, $U_{il}(\theta_{il}, \phi_{il}, \lambda_{il})$ represents the unitary operation acting on the $i$-th qubit in the $l$-th layer. Let $|\psi_o\rangle$ represent the quantum state of the circuit after the unitary operations have been applied. Our decoding process is described by the following equations:

$$E_i = \langle \psi_o | Z_i | \psi_o \rangle \tag{5}$$

$$Z_i = I^{\otimes(i-1)} \otimes Z \otimes I^{\otimes(n-i)} \tag{6}$$

$$Z = \begin{pmatrix} 1 & 0 \\ 0 & -1 \end{pmatrix} \tag{7}$$

$$I = \begin{pmatrix} 1 & 0 \\ 0 & 1 \end{pmatrix} \qquad (8)$$

We measure each qubit in the circuit using Equation (5), where $E_i$ represents the expectation value and $Z_i$ is the observable for the $i$-th qubit. Although the circuit or filter's operations and calculations occur in the complex domain, the resulting expectation values are real and fall within the range of $[-1, 1]$. For instance, an expectation value close to $-1$ suggests a high probability of the qubit being in the $|1\rangle$ state and vice versa.

Each expectation value is mapped to a distinct output channel. A filter with $n$ qubits will produce $n$ feature maps, and the total number of filters required is $\lfloor \frac{C_{\text{out}}+n-1}{n} \rfloor$. However, if the total number of feature maps exceeds the user-defined number of output channels, we discard the extra maps, retaining only $C_{out}$ feature maps. Finally, we add a bias term to the generated features for each output channel, just like in a classical convolutional layer.

## 2.2 FQN

The FQN architecture is designed to be parameter-efficient, built exclusively with Quanv1D layers, as shown in Fig. 1. It consists of three primary components: embedding, propagation, and projection. In summary, the embedding layer transforms the input into a high-dimensional feature space, the propagation layers employ dilation (Yu, 2015) to iteratively improve feature representations with longer receptive fields, and the projection layer transfers these enhanced features to a desired output space.

The embedding layer expands the raw input data into a learned embedding dimension, $dim_e$, which helps aggregate local information and prepares the data for more complex hierarchical feature extraction. For long sequences, the kernel size $k_e$ can remain large without shortening the sequence length, as padding is automatically adjusted using $\lfloor \frac{k_e}{2} \rfloor$. However, this adjustment is only applicable to kernels with odd sizes. In addition, strides can reduce the input sequence length if necessary. For example, setting $s_e = 3$ will shorten the embedded feature sequence to one-third of the original input length.

After the embedding stage, the propagation layers form the core of the architecture. There are $depth$ propagation layers, each composed of a Quanv1D layer, followed by batch normalization and ReLU activation. These layers improve the embedded representation over time by reducing the internal covariate shift and adding nonlinearity to the data. The dilation rate, $d_i = i$ (where $i$ is the layer index), linearly increases the network's receptive field as the depth grows. This allows the model to capture dependencies at multiple scales–from local interactions in the lower layers to long-range dependencies in the deeper layers–without increasing the number of parameters. In the propagation layers, no padding was applied, and $s_{prop}$ was set to one. However, the kernel size $k_{prop}$ was adjusted based on the specific dataset.

Following the propagation layers, the projection layer maps the multi-scale, transformed features to a space corresponding to the target classes. This allows the network to produce class-specific feature maps. For this layer, the kernel size is set to $k_{proj} = 1$, with a stride of $s_{proj} = 1$, no padding ($p_{proj} = 0$), and a dilation rate of $d_{proj} = 1$. We then apply global average pooling across the channel dimension and pass the pooled features to a softmax classifier for final prediction. This approach helps prevent overfitting by reducing the number of trainable parameters and serves as a form of regularization.

## 3 EXPERIMENTS

### 3.1 DATASETS AND MODELS

To evaluate the performance of FQN (and Quanv1D), we utilized 20 datasets from the UEA and UCR time series archives (Bagnall et al., 2018; Dau et al., 2019). We randomly selected the datasets but ensured a broad spectrum of practical applications. As such, the selected binary and multi-class datasets cover 15 different fields, with inputs that have different sizes (up to 64 channels) and lengths (up to 18530). Table 3 provides a detailed description of these datasets, and Appendix C outlines the selection process used in this study.

Table 1: Classification performance across datasets. Each experiment is repeated five times. We present the mean and standard deviation for test accuracy and provide a ratio for trainable parameter count, which illustrates how each model's count compares to the FQN. Bold indicates the best performance.

| Dataset | Test accuracy (%) | | | | Parameters (ratio-to-FQN) | | | |
|---|---|---|---|---|---|---|---|---|
| | FQN | FCN | ModernTCN | QuanvNet* | FQN | FCN | ModernTCN | QuanvNet* |
| D1 | 96.4 (1.2) | 96.4 (0.8) | **97.8** (1.2) | 93.2 (5.1) | ×**1** | ×4.0 | ×4.7 | ×108.0 |
| D2 | 59.5 (1.0) | 59.0 (1.1) | **62.5** (2.2) | 58.0 (2.6) | ×**1** | ×3.0 | ×5.7 | ×101.8 |
| D3 | 98.7 (0.7) | **98.8** (0.5) | 96.8 (1.4) | 96.0 (2.0) | ×**1** | ×9.8 | ×11.1 | ×31.6 |
| D4 | 76.5 (1.2) | 80.9 (0.3) | **82.6** (1.3) | 70.9 (1.8) | ×**1** | ×5.2 | ×5.3 | ×35.9 |
| D5 | **80.5** (4.1) | 74.5 (2.7) | 80.0 (3.5) | 76.0 (7.2) | ×**1** | ×2.1 | ×3.5 | ×51.1 |
| D6 | 93.3 (1.2) | **99.2** (0.8) | 98.9 (1.2) | 87.2 (3.3) | ×**1** | ×6.8 | ×4.0 | ×20.8 |
| D7 | **95.8** (1.7) | 90.3 (3.3) | 80.0 (2.7) | 81.8 (8.6) | ×**1** | ×4.9 | ×6.9 | ×28.9 |
| D8 | 98.1 (1.3) | **100.0** (0.0) | 99.7 (0.7) | 89.1 (5.3) | ×**1** | ×6.4 | ×5.0 | ×15.2 |
| D9 | **75.7** (2.3) | 73.3 (4.5) | 71.5 (1.4) | 66.5 (12.3) | ×**1** | ×7.9 | ×12.2 | ×27.2 |
| D10 | **100.0** (0.0) | **100.0** (0.0) | **100.0** (0.0) | **100.0** (0.0) | ×**1** | ×4.4 | ×11.5 | ×33.2 |
| D11 | 95.1 (0.4) | **97.8** (0.0) | 88.8 (4.6) | 95.6 (0.6) | ×**1** | ×7.9 | ×31.4 | ×27.3 |
| D12 | **99.5** (1.1) | **99.5** (1.1) | 59.0 (5.3) | 98.5 (1.1) | ×**1** | ×3.4 | ×21.9 | ×27.0 |
| D13 | 72.7 (1.1) | **74.1** (3.4) | 64.1 (3.8) | 73.7 (4.1) | ×**1** | ×2.1 | ×255.5 | ×79.3 |
| D14 | **99.3** (1.0) | 98.9 (1.0) | 96.0 (3.5) | – | ×**1** | ×11.1 | ×38.7 | – |
| D15 | 29.1 (1.7) | 28.6 (2.0) | **29.7** (5.4) | – | ×**1** | ×6.6 | ×107.3 | – |
| D16 | 97.8 (0.4) | **99.9** (0.2) | 99.6 (0.2) | – | ×**1** | ×5.2 | ×33.5 | – |
| D17 | **56.8** (2.4) | 44.7 (0.9) | 52.9 (5.5) | – | ×**1** | ×7.2 | ×118.8 | – |
| D18 | 34.5 (3.5) | 31.1 (3.2) | **36.6** (5.9) | – | ×**1** | ×2.6 | ×96.8 | – |
| D19 | **56.1** (2.8) | 55.7 (2.9) | 55.4 (2.8) | – | ×**1** | ×7.6 | ×248.8 | – |
| D20 | 52.4 (1.1) | 51.1 (6.6) | **55.0** (2.4) | – | ×**1** | ×10.6 | ×12600.9 | – |
| Average | **78.4** | 77.7 | 75.3 | – | ×**1** | ×6.0 | ×681.2 | – |

*The results for QuanvNet are incomplete because it can handle univariate data only.

Our goal was to compare FQN to a fully convolutional network (FCN) that had the same architecture and hyperparameters but contained classical convolutional layers. This comparison helped us examine the differences between Quanv1D and Conv1D in terms of training and inference. To assess FQN's benchmarking potential, we included ModernTCN, the current state-of-the-art (SOTA) in time series classification, in the comparison (Luo & Wang, 2024). We also added the latest 1D quanvolutional network, QuanvNet, proposed by Rivera-Ruiz et al. (2023).

## 3.2 TIME SERIES CLASSIFICATION

According to the performance results outlined in Table 1, FQN is comparable to ModernTCN and FCN, even outperforming them in certain cases, despite having substantially fewer parameters. While FQN and FCN both outperformed other models in eight cases, resulting in a tie, FQN was better overall. It achieved a higher average with the least trainable parameters. Since our goal with FQN was to develop a model comparable or equivalent to FCN, the results in Table 1 highlight the potential of quantum-based representation learning.

Interestingly, FQN outperformed the current SOTA model, ModernTCN. The input's length and dimension largely determine ModernTCN's parameter count, primarily because of its proposed embedding layer and the fully connected layers in its classifier head. Although it is generally believed that more parameters lead to better performance, a model with sufficient parameters to properly fit the data—provided it is well-suited to the task—should not underperform compared to a model with significantly more parameters. As shown in Table 5, ModernTCN clearly exhibits overfitting in multiple datasets.

Across all datasets, FQN consistently outperforms QuanvNet. From a stability standpoint, QuanvNet shows the greatest variance in accuracy. Its subpar performance can largely be attributed to two key factors: (i) the constraints imposed by a fixed, small kernel length, which limits its ability to capture temporal relationships effectively, and (ii) the reliance on a single circuit as a filter, which hinders its capacity to extract diverse patterns. Additionally, it can only process univariate time series data. As such, we were unable to evaluate QuanvNet across all 20 datasets.

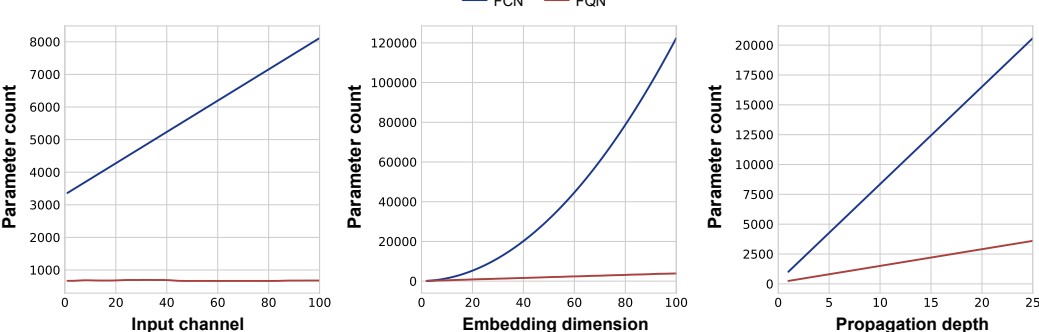

Figure 2: Variation of model parameter count against hyperparameter change. The base hyperparameters are: $C_{in} = 3$, $k_e = 3$, $s_e = 2$, $dim_e = 16$, $k_{prop} = 3$, and $depth = 4$. The parameter change is then observed by varying one hyperparameter at a time. For illustration, we used a Gaussian filter for line smoothing.

### 3.3 PARAMETER EFFICIENCY

Although both models share the same architecture and hyperparameters, FCN is, on average, six times lighter than FQN. Figure 2 illustrates the parameter differences between the two models across various hyperparameter settings. As seen in the graph, the primary difference comes from $dim_e$, where FCN's parameter count grows exponentially, while FQN shows only a gradual linear increase. While both models experience linear scaling in parameter count as $depth$ increases, the rate of increase is significantly higher for FCN. Additionally, Quanv1D's efficient encoding and decoding processes help keep FQN's parameter count nearly constant for increasing $c_{in}$, in contrast to FCN's steady linear growth. A higher $C_{in}$ typically requires more qubits (and hence, more parameters), but Quanv1D minimizes this by utilizing all available qubits in its circuits, thereby reducing the total number of filters needed. This synergistic approach is what makes Quanv1D (and FQN) more efficient in managing parameters.

The results in Table 2 show that FQN consistently maintains comparable losses across the training, validation, and test sets. This is in contrast to its classical counterpart, FCN, which often overfits and achieves almost zero training loss but much higher validation and test losses. However, after closely looking at Table 2 and how FQN behaves during training, we hypothesize that a higher number of parameters is not the sole reason why FCN fell behind FQN in multiple datasets. To test this hypothesis, we rerun the experiments, adjusting the hyperparameters of FCN to ensure that the trainable parameter for both FCN and FQN is equal. While reducing parameters lessened overfitting in six cases (dropping from 13 to seven), the test accuracy also declined for ten cases. This goes to show that FQN offers more stable learning when compared to FCN, despite the number of parameters used.

### 3.4 SELF-REGULARIZATION

Table 2 provides empirical evidence of FQN's self-regularization, which enabled it to generalize more effectively in most test cases. Although this implicit regularization is partly due to the reduced number of parameters, we believe it also stems from the fundamental nature of quantum operations within the Quanv1D layer. To illustrate this, consider a single-wire quanvolutional kernel (or quantum circuit) with a single unitary operator. The derivatives of the output, $Q_{out}$, with respect to $\theta$ and $\lambda$ for a given patch are expressed as follows:

$$\frac{dQ_{out}}{d\theta} = -\pi(x_1^2 - x_2^2)\sin(\pi\theta) - 2\pi x_1 x_2 \cos(\lambda)\cos(\theta\pi) \tag{9}$$

$$\frac{dQ_{out}}{d\lambda} = 2x_1 x_2 \sin(\lambda)\sin(\theta\pi) \tag{10}$$

Table 2: The mean loss values of FQN and FCN across datasets with different splits. For FCN*, the total number of trainable parameters matches that of FQN.

| Dataset | FQN | | | FCN | | | | FCN* | | | | FCN-FCN* |
| | Train | Val | Test | Train | Val | Test | Overfitting? | Train | Val | Test | Overfitting? | Change in accuracy (%) |
|---|---|---|---|---|---|---|---|---|---|---|---|---|
| D1 | 0.27 | 0.29 | 0.31 | 0.00 | 0.01 | 0.23 | ✓ | 0.00 | 0.01 | 0.17 | ✓ | 0.5 |
| D2 | 0.61 | 0.68 | 0.68 | 0.01 | 2.87 | 2.51 | ✓ | 0.03 | 3.21 | 2.96 | ✓ | 0.5 |
| D3 | 0.86 | 0.90 | 0.88 | 0.00 | 0.05 | 0.05 | ✗ | 0.00 | 0.03 | 0.02 | ✗ | 0.3 |
| D4 | 0.55 | 0.59 | 0.57 | 0.01 | 0.92 | 0.81 | ✓ | 0.04 | 0.64 | 0.72 | ✓ | -2.3 |
| D5 | 0.43 | 0.60 | 0.55 | 0.00 | 1.29 | 1.36 | ✓ | 0.01 | 1.09 | 0.59 | ✓ | 6.5 |
| D6 | 0.40 | 0.44 | 0.43 | 0.00 | 0.00 | 0.02 | ✗ | 0.00 | 0.02 | 0.02 | ✗ | 0.0 |
| D7 | 0.38 | 0.39 | 0.41 | 0.02 | 0.26 | 0.29 | ✓ | 0.13 | 0.48 | 0.27 | ✗ | 1.2 |
| D8 | 0.68 | 0.54 | 0.59 | 0.17 | 0.00 | 0.00 | ✗ | 0.17 | 0.08 | 0.08 | ✗ | -2.2 |
| D9 | 0.48 | 0.82 | 0.57 | 0.00 | 4.19 | 1.70 | ✓ | 0.00 | 3.10 | 1.38 | ✓ | 0.7 |
| D10 | 0.66 | 0.64 | 0.68 | 0.02 | 0.01 | 0.01 | ✗ | 0.13 | 0.08 | 0.07 | ✗ | 0.0 |
| D11 | 0.38 | 0.42 | 0.39 | 0.00 | 0.36 | 0.15 | ✓ | 0.00 | 0.32 | 0.12 | ✓ | -0.3 |
| D12 | 0.57 | 0.57 | 0.56 | 0.02 | 0.07 | 0.03 | ✗ | 0.44 | 0.68 | 0.34 | ✗ | -2.4 |
| D13 | 0.58 | 0.67 | 0.58 | 0.01 | 2.14 | 1.19 | ✓ | 0.55 | 0.72 | 0.57 | ✗ | -3.2 |
| D14 | 0.50 | 0.53 | 0.51 | 0.00 | 0.07 | 0.03 | ✗ | 0.01 | 0.08 | 0.07 | ✗ | -2.2 |
| D15 | 1.31 | 1.43 | 1.38 | 0.32 | 2.36 | 2.83 | ✓ | 0.71 | 1.94 | 1.97 | ✓ | 5.5 |
| D16 | 0.36 | 0.37 | 0.37 | 0.00 | 0.01 | 0.00 | ✗ | 0.00 | 0.00 | 0.01 | ✗ | -0.3 |
| D17 | 0.61 | 0.71 | 0.70 | 0.41 | 1.07 | 1.09 | ✓ | 0.64 | 0.75 | 0.71 | ✗ | 7.6 |
| D18 | 1.28 | 1.40 | 1.35 | 0.41 | 1.79 | 1.69 | ✓ | 1.32 | 1.41 | 1.39 | ✗ | -5.1 |
| D19 | 0.55 | 0.75 | 0.69 | 0.07 | 1.85 | 1.74 | ✓ | 0.61 | 0.85 | 0.75 | ✗ | -2.9 |
| D20 | 0.67 | 0.71 | 0.70 | 0.39 | 1.12 | 1.14 | ✓ | 0.68 | 0.70 | 0.71 | ✗ | -3.7 |

Both equations demonstrate a sinusoidal relationship between the derivatives and the weight values: $\theta$ and $\lambda$. A sinusoidal function, naturally bound between -1 and 1, inherently constrains the gradient update due to its periodicity. Additionally, the amplitude embedding restricts the input values to a range between 0 and 1. As the inputs are the coefficients of the sinusoids, the input scaling limits the magnitude of the gradient updates even further. Together, these factors help prevent extreme gradient values, promoting self-regularization.

In a similar way, we can extend this illustration to a two-qubit, two-unitary circuit. The gradients for each wire with respect to the weights are as follows:

$$\frac{dQ_{out}}{d\theta_1} = -\pi(x_1^2 + x_2^2 - x_3^2 - x_4^2)\sin(\theta_1\pi) - 2\pi(x_1x_3 + x_2x_4)\cos(\lambda_1)\cos(\pi\theta_1) \tag{11}$$

$$\frac{dQ_{out}}{d\theta_2} = -\pi(x_1^2 - x_2^2 + x_3^2 - x_4^2)\sin(\theta_2\pi) - 2\pi(x_1x_2 + x_3x_4)\cos(\lambda_2)\cos(\pi\theta_2) \tag{12}$$

$$\frac{dQ_{out}}{d\lambda_1} = (x_1x_3\sin(\lambda_1) + x_2x_4\sin(\lambda_1))\sin(\pi\theta_1) \tag{13}$$

$$\frac{dQ_{out}}{d\lambda_2} = (x_1x_2\sin(\lambda_2) + x_3x_4\sin(\lambda_2))\sin(\pi\theta_2) \tag{14}$$

$\theta_i$ and $\lambda_i$ represent the weights of the $i$-th wire. The regularization effect is also evident in Equations (11, 12, 13, and 14). The gradient update value discussed here applies to a single layer, but because the model is fully quanvolutional, adding more layers won't affect this behavior. Since gradient updates occur via a multiplicative chain rule, a similar regularization effect will propagate across all layers. The derivation for the single-wire circuit can be found in Appendix F, and the equations for the two-wire circuit were derived using SymPy (Meurer et al., 2017). Note that the derivatives with respect to $\phi$ are always zero.

### 3.5 IMPACT OF FINITE SHOTS

Despite FQN's promise for classifying real-world data, its application on actual quantum computers remains a challenge. Quanv1D is mostly theoretical, as we rely on analytical or raw expectation values. In practice, this approach is not feasible because expectation values must be measured using a finite number of shots. To assess our model's performance under such conditions, we conducted

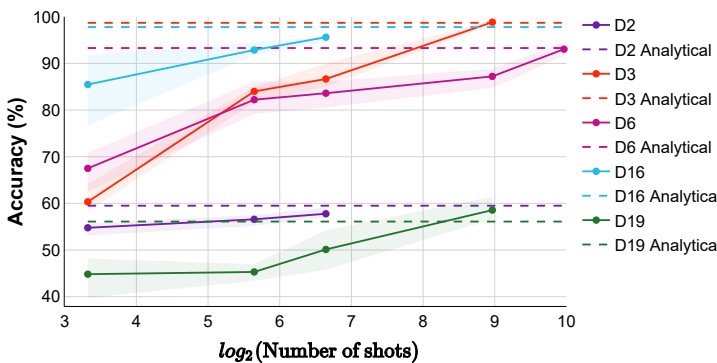

Figure 3: Accuracy vs. number of shots. On the test sets, we evaluated the model five times for each shot count. Dashed lines indicate the analytical results, while the solid line represents the mean accuracy of the shot-based models. The shaded area around the solid line reflects the uncertainty range, spanning from the minimum to maximum accuracy for each shot count.

experiments starting with ten shots and continued until FQN achieved accuracy within 3% of the analytical benchmark. Due to the high computational cost of the simulation, we had to limit our analysis to this threshold. Nevertheless, since the mean value obtained from the shots is a sample mean rather than the theoretical expected value, the law of large numbers (Evans & Rosenthal, 2004) predicts that as the number of shots increases significantly, the sample mean will eventually converge to the true expected value. The results are shown in Fig. 3. For these tests, we randomly selected five datasets from a pool of 20.

The number of shots needed to achieve the desired accuracy may differ depending on the dataset. However, a consistent trend is that FQN exhibits erratic behavior at very low shot counts due to randomness. This instability diminishes as the number of shots increases, eventually leading to stabilization. Also, the introduced noise can sometimes enhance performance, even exceeding the noise-free analytical baseline, as observed with D19. This outcome aligns with earlier studies demonstrating the robustness of QML to noise (Cross et al., 2015; Du et al., 2021), which show that noise can actually aid in data learning tasks. Consequently, in real-world scenarios where noise is unavoidable, FQN has the potential to become more generalized and error-tolerant, when applied to practical data.

## 4    CONCLUDING REMARKS

The primary objective of this study was to introduce a QML algorithm for quantum computers, aiming to create a quantum equivalent of a well-established and widely used classical method. We designed our proposed quanvolution algorithm, an analog to convolution, with NISQ-related constraints in mind, and it proved effective in temporal data learning tasks. In fact, it was able to outperform contemporary classical models in some cases, thanks to its efficient parameter management and inherent regularization. Despite our model being a theoretical framework, it remained effective when we simulated a realistic scenario with statistical noise. Our proposed model, FQN, incorporates non-linear activation functions, which present a challenge in a quantum environment. But the computational overhead for these activation functions is negligible, and the network should, in theory, work well in a classical-quantum hybrid framework. Previous studies have demonstrated promising results for similar hybrid approaches with smaller models (Tacchino et al., 2019; 2020). As the world moves toward fault-tolerant quantum computing with increased qubits, we are optimistic that, with the right settings, our model will perform similarly. Nonetheless, significant progress is still required before reaching that point.

### 4.1    LIMITATIONS

For this study, we tested our FQN only on classification tasks and have yet to explore regression and data imputation problems. Using a fully quanvolutional structure for these tasks will be challenging,

as Quanv1D's output is in the range of -1 to 1, which limits its applicability for tasks that require output values beyond this range. Additionally, for classification problems, we noticed that training FQN gets difficult when the number of classes exceeds 10, either due to the limited number of parameters or the usage of classical optimizers on a quantum model. Also, we could not determine the true operating range of FQN because we only tested it on a subset of the UEA and UCR repositories and not all the datasets, as the authors suggested (Dau et al., 2019).

In addition to these limitations, we believe it is important to address the constraints related to hardware implementation. Our quanvolutional layer is largely theoretical, simulated using classical computing. Given the constraints associated with the NISQ era, it remains uncertain whether we can achieve this level of implementation, as we did not have access to real hardware to verify our algorithm. While both Quanv1D and FQN showed strong performance in tasks involving statistical noise, their robustness against other challenges–such as decoherence, environmental noise, error accumulation, and gate fidelity–has yet to be demonstrated. Moreover, although amplitude encoding offers improved parameter efficiency for the model, it significantly increases circuit depth (Kharsa et al., 2023; Schuld, 2018), posing further challenges. Finally, the high computational complexity of FQN makes classical simulation difficult. For comparison, the time complexity of quanvolution is $\mathcal{O}(k^2 \times c_{in}^2 \times c_{out})$, whereas that of convolution is $\mathcal{O}(k \times c_{in} \times c_{out})$, where $k$ is the kernel length, $c_{in}$ is the input dimension, and $c_{out}$ is the number of output channels. Optimizing the design to reduce complexity remains an area for future exploration.

### 4.2 FUTURE WORK

To address the current limitations, our immediate focus will be on improving the model's performance when handling more than 10 classes and optimizing the overall design to reduce time complexity. Following this, we plan to benchmark the model using the full UEA and UCR repositories Dau et al. (2019); Bagnall et al. (2018), which will help establish the operational range of quanvolution for classification tasks. After completing the classification phase, we will address the range limitation of -1 to 1. While the output range of the quantum layer cannot be altered, we aim to design an activation function that maps outputs to a higher range while preserving the distinct quantum properties. Achieving this mapping could open new avenues for exploration.

Before advancing to our ultimate goal of developing a 2D quanvolution, we aim to introduce explainability to the current 1D quanvolution framework. Although inspired by classical 1D convolution, the quantum operations introduce complexities that make explainability less intuitive. Understanding the mathematical principles underlying these processes will not only enhance interpretability but may also offer insights for further design improvements. Transitioning from 1D to 2D quanvolution while maintaining this level of modularity will be a significant challenge, but we believe this progression is the logical next step, especially for expanding applications into the domains of image processing and computer vision.

### REPRODUCIBILITY STATEMENT

The main text and the appendices provide detailed instructions for building the models and replicating the results. After the peer-review process, we will share the code in a public repository.

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

## A  PRELIMINARIES

Quantum bits, or qubits, are the fundamental units of quantum computation. Unlike classical bits, which can only exist in one of two definite states (0 or 1), qubits can be in the 0 state ($|0\rangle$), the 1 state ($|1\rangle$), or any linear combination (superposition) of these states, expressed as $|\psi\rangle = \alpha |0\rangle + \beta |1\rangle$. Here, $\alpha$ and $\beta$ are complex numbers that satisfy the condition $|\alpha|^2 + |\beta|^2 = 1$. This superposition property allows quantum operations to act on multiple states simultaneously, something that

Table 3: Descriptions of the datasets used in the study.

| Code | Dataset | Sample | Length | Dim | Type | Domain |
|------|---------|--------|--------|-----|------|--------|
| D1 | Chinatown | 363 | 24 | 1 | Traffic | Urban planning |
| D2 | SharePriceIncrease | 1931 | 60 | 1 | Financial | Stock market |
| D3 | SyntheticControl | 600 | 60 | 1 | Simulated | Synthetic data analysis |
| D4 | PhalangesOutlinesCorrect | 2658 | 80 | 1 | Image | Osteology |
| D5 | ECG200 | 200 | 96 | 1 | ECG | Cardiovascular diagnostics |
| D6 | PowerCons | 360 | 144 | 1 | Device | Smart grid |
| D7 | ToeSegmentation2 | 166 | 343 | 1 | Motion | Biomechanics |
| D8 | DiatomSizeReduction | 322 | 345 | 1 | Image | Microbiology |
| D9 | Earthquakes | 461 | 512 | 1 | Sensor | Seismology |
| D10 | InsectEPGRegularTrain | 311 | 601 | 1 | EPG | Entomology |
| D11 | StarLightCurves | 9236 | 1024 | 1 | Sensor | Astronomy |
| D12 | NerveDamage | 204 | 1500 | 1 | EMG | Neurology |
| D13 | BinaryHeartbeat | 409 | 18530 | 1 | Audio | Cardiovascular diagnostics |
| D14 | Epilepsy | 275 | 206 | 3 | Sensor | Human activity recognition |
| D15 | EthanolConcentration | 524 | 1751 | 3 | Spectroscopy | Chemical analysis |
| D16 | Blink | 950 | 510 | 4 | EEG | Brain-computer interface |
| D17 | SelfRegulationSCP2 | 380 | 1152 | 7 | EEG | Brain-computer interface |
| D18 | HandMovementDirection | 234 | 400 | 10 | EEG | Brain-computer interface |
| D19 | FingerMovements | 416 | 50 | 28 | EEG | Brain-computer interface |
| D20 | MotorImagery | 378 | 3000 | 64 | EEG | Brain-computer interface |

ECG: electrocardiogram; EPG: electrical penetration graph; EMG: electromyogram; EEG: electroencephalogram

classical computers cannot achieve. When combined with the phenomena of entanglement and interference, superposition forms the basis of "quantum advantage"–the ability of quantum systems to solve specific problems more efficiently than classical systems.

Quantum circuits provide the framework for modeling quantum computations; wires represent qubits, and gates correspond to quantum operations that manipulate these qubits. In the context of applying QML to classical data, quantum circuits typically consist of three main phases: encoding, manipulation, and measurement.

1. **Encoding:** In this phase, classical information is mapped into a quantum Hilbert space, preparing the quantum system for computation. Two widely used encoding methods are angle encoding and amplitude encoding. Angle encoding represents classical values as angles of rotation gates applied to qubits, typically starting from the $|0\rangle$ state. This method requires $n$ qubits to encode $n$ classical features. Amplitude encoding or embedding, in contrast, stores classical data in the amplitudes of a quantum state, allowing $n$ qubits to represent up to $2^n$ classical values classical values. While amplitude encoding is highly efficient in terms of qubit usage, preparing such states can be computationally intensive.

2. **Manipulation:** After encoding, the quantum data is processed using a sequence of quantum gates. Researchers often design parameterized quantum circuits, known as ansatz, which include gates such as Hadamard, Controlled-NOT (CNOT), and rotations around the X, Y, and Z axes. These gates enable critical quantum phenomena for quantum advantage. For example, the Hadamard gate creates superposition, while the CNOT gate establishes entanglement.

3. **Measurement:** The final stage involves measuring quantum states to extract classical outcomes. Measurement causes the quantum state to collapse into one of its basis states (either 0 or 1). Due to the probabilistic nature of quantum measurement, multiple repetitions (referred to as "shots") are performed to gather sufficient statistics. With enough measurements or experimental reruns, the outcomes reliably reflect the expected or analytical results of the quantum computation.

# B EXPERIMENTAL SETUP

In this study, we conducted the experiments on an NVIDIA GeForce RTX 4060 Ti 16GB GPU using PyTorch (Paszke et al., 2019).

Table 4: Total trainable parameters of FQN with its hyperparameters.

| Dataset | Batch size | Parameters | Decay factor | $k_e$ | $s_e$ | $dim_e$ | $k_{prop}$ | $depth$ |
|---------|-----------|-----------|-------------|-------|-------|---------|-----------|---------|
| D1 | 64 | 856 | 0.85 | 9 | 1 | 16 | 3 | 4 |
| D2 | 256 | 908 | 0.85 | 15 | 3 | 16 | 3 | 3 |
| D3 | 128 | 2988 | 0.85 | 15 | 2 | 48 | 3 | 4 |
| D4 | 256 | 2576 | 0.75 | 21 | 2 | 32 | 3 | 4 |
| D5 | 32 | 1808 | 0.75 | 31 | 4 | 16 | 3 | 4 |
| D6 | 64 | 4434 | 0.75 | 15 | 3 | 32 | 5 | 4 |
| D7 | 32 | 3200 | 0.75 | 31 | 5 | 24 | 5 | 5 |
| D8 | 32 | 6098 | 0.8 | 45 | 4 | 32 | 9 | 4 |
| D9 | 64 | 3400 | 0.75 | 45 | 8 | 24 | 7 | 4 |
| D10 | 32 | 2796 | 0.75 | 33 | 5 | 16 | 9 | 5 |
| D11 | 128 | 3400 | 0.8 | 21 | 8 | 32 | 5 | 5 |
| D12 | 32 | 3440 | 0.85 | 41 | 10 | 16 | 7 | 6 |
| D13 | 32 | 1166 | 0.9 | 15 | 20 | 8 | 7 | 5 |
| D14 | 64 | 6072 | 0.85 | 25 | 3 | 64 | 3 | 5 |
| D15 | 128 | 4858 | 0.85 | 45 | 10 | 24 | 7 | 7 |
| D16 | 128 | 3618 | 0.85 | 41 | 5 | 24 | 5 | 5 |
| D17 | 64 | 4882 | 0.85 | 41 | 9 | 32 | 5 | 5 |
| D18 | 128 | 1488 | 0.75 | 9 | 2 | 8 | 5 | 9 |
| D19 | 64 | 728 | 0.75 | 5 | 2 | 16 | 3 | 4 |
| D20 | 16 | 524 | 0.9 | 3 | 25 | 16 | 3 | 3 |

## C  DATASET SETUP

The datasets used in this experiment were taken from the UEA and UCR archives (Bagnall et al., 2018; Dau et al., 2019). Among the 150+ datasets available in the archive, we have considered a subset of 20. The selected datasets along with their descriptions are provided in Table 3. Our dataset selection criteria were primarily based on the domain and its real-life applicability, i.e., we wanted to cover as many domains as possible to test our proposed model. The considered datasets come from 15 different domains and include a mix of both binary and multi-class classification tasks. A major portion of the selected datasets are univariate in nature, as QuanvNet (Rivera-Ruiz et al., 2023), one of our baselines, only works on univariate data. One exception to the criteria was the selection of multiple EEG datasets. This was due to the high variability and low statistical power of EEG datasets (Davoudi et al., 2023; Button et al., 2013), along with their high dimensionality.

All datasets were processed following the same procedure. We first applied Z-normalization to the entire dataset, followed by a 60:20:20 split for training, validation, and testing. For multi-dimensional data, we performed Z-normalization on each input channel individually. We saved the normalization factors and data splits to ensure reproducibility and fair comparisons among models trained on the same datasets. Additionally, we calculated class weights and integrated them into the cross-entropy loss function to address class imbalances in several datasets.

## D  TRAINING SETUP

We trained each model for 200 epochs, opting for a learning rate scheduler instead of early stopping. As mentioned before, our objective was to evaluate the differences in learning between Quanv1D and traditional convolutional layers. The scheduler had a patience of 5 epochs and a relative threshold of 0.001, monitoring the validation loss to reduce the learning rate by different factors when performance plateaued. We varied the reduction factor to ensure stable training and minimize overfitting. However, for consistency, the reduction factor remained the same across all comparison models for each dataset. The data specific reduction factors are presented in Table 4.

We optimized all the models using Adam (Kingma, 2014), starting with default values set by PyTorch. The initial learning rate for all datasets was 0.01, except for D10 and D12, where it was 0.001. We repeated all the experiments five times without using any specific seeds.

# E   MODEL SETUP

Table 4 presents FQN's hyperparameters for each dataset. Note that, for fairness, we also use the same batch size for other models under comparison. In the case of FCN, the hyperparameters are identical to the values presented in Table 4. In the table, $depth$ refers to the number of propagation layers.

We utilized the official implementation of ModernTCN from `https://github.com/luodhhh/ModernTCN`, retaining the hyperparameters recommended by Luo & Wang (2024) for time series classification. The only modification was to adjust the embedding dimension to align with FQN and FCN, ensuring a fair comparison. For QuanvNet, we implemented it from scratch based on the approach outlined by Rivera-Ruiz et al. (2023).

# F   GRADIENT DERIVATION

Let us consider a one-wire, one-unitary quantum circuit where the number of inputs is 2. Before being processed by the circuit, the inputs undergo amplitude embedding, such that the input vector is given by:

$$|x_{input}\rangle = \begin{pmatrix} x_1 \\ x_2 \end{pmatrix} \tag{15}$$

The output state of the quantum circuit after applying a unitary transformation parameterized by angles $\theta$, $\phi$, and $\lambda$ is described as:

$$|\psi_o\rangle = U(\theta, \phi, \lambda) * x_{input} = \begin{pmatrix} x_1 * \cos(\theta\frac{\pi}{2}) - x_2 * e^{i\lambda}\sin(\theta\frac{\pi}{2}) \\ x_1 * e^{i\phi}\sin(\theta\frac{\pi}{2}) + x_2 * e^{i(\phi+\lambda)}\cos(\theta\frac{\pi}{2}) \end{pmatrix} \tag{16}$$

After passing through the measurement block, the output $Q_{out}$ is computed as the expectation value of the Pauli-Z operator:

$$Q_{out} = \langle\psi_o|Z|\psi_o\rangle \tag{17}$$

Substituting $|\psi_o\rangle$ into this expression and simplifying, we obtain:

$$Q_{out} = x_1^2 \cos^2(\theta\frac{\pi}{2}) + x_2^2 \sin^2(\theta\frac{\pi}{2}) - x_1 x_2 \cos\lambda\sin(\pi\theta)$$
$$-x_1^2 \sin^2(\theta\frac{\pi}{2}) - x_2^2 \cos^2(\theta\frac{\pi}{2}) - x_1 x_2 \cos\lambda\sin(\pi\theta) \tag{18}$$
$$= (x_1^2 - x_2^2)\cos^2(\theta\frac{\pi}{2}) - (x_1^2 - x_2^2)\sin^2(\theta\frac{\pi}{2}) - 2x_1 x_2 \cos\lambda\sin(\pi\theta)$$

Now, the derivatives of $Q_{out}$ with respect to $\theta$, $\phi$, and $\lambda$ are:

$$\frac{dQ_{out}}{d\theta} = -2\frac{\pi}{2}(x_1^2 - x_2^2)\cos(\theta\frac{\pi}{2})\sin(\theta\frac{\pi}{2})$$
$$-2\frac{\pi}{2}(x_1^2 - x_2^2)\sin(\theta\frac{\pi}{2})\cos(\theta\frac{\pi}{2}) - 2\pi x_1 x_2 \cos(\lambda)\sin(\pi\theta) \tag{19}$$
$$= -\pi(x_1^2 - x_2^2)\sin(\pi\theta) - 2\pi x_1 x_2 \cos(\lambda)cos(\pi\theta)$$

$$\frac{dQ_{out}}{d\lambda} = 2x_1 x_2 \sin(\lambda)\sin(\pi\lambda) \tag{20}$$

$$\frac{dQ_{out}}{d\phi} = 0 \tag{21}$$

## G  PARAMETER EFFICIENCY (EXTENDED)

Table 5 highlights the tendency of ModernTCN to overfit. Out of the 20 datasets evaluated, ModernTCN overfitted 14 of them, achieving near-zero training loss but exhibiting significantly higher test loss. Unlike FCN, ModernTCN's rigid model structure precluded the exploration of reducing parameter counts for overfitting mitigation. However, our FCN-based exploration demonstrates that FQN's design and self-regularization will give it a competitive edge over ModernTCN, even though parameter reduction in ModernTCN can improve some overfitting.

Table 5: The mean loss values of FQN and ModernTCN across datasets with different splits.

| Dataset | FQN | | | ModernTCN | | | Overfitting? |
|---|---|---|---|---|---|---|---|
| | Train | Val | Test | Train | Val | Test | |
| D1 | 0.27 | 0.29 | 0.31 | 0.00 | 0.01 | 0.20 | ✓ |
| D2 | 0.61 | 0.68 | 0.68 | 0.23 | 1.31 | 1.19 | ✓ |
| D3 | 0.86 | 0.90 | 0.88 | 0.00 | 0.16 | 0.14 | × |
| D4 | 0.55 | 0.59 | 0.57 | 0.18 | 0.52 | 0.51 | × |
| D5 | 0.43 | 0.60 | 0.55 | 0.00 | 1.21 | 1.19 | ✓ |
| D6 | 0.40 | 0.44 | 0.43 | 0.00 | 0.00 | 0.04 | × |
| D7 | 0.38 | 0.39 | 0.41 | 0.00 | 2.06 | 3.35 | ✓ |
| D8 | 0.68 | 0.54 | 0.59 | 0.00 | 0.01 | 0.01 | × |
| D9 | 0.48 | 0.82 | 0.57 | 0.00 | 12.73 | 3.67 | ✓ |
| D10 | 0.66 | 0.64 | 0.68 | 0.01 | 0.00 | 0.00 | × |
| D11 | 0.38 | 0.42 | 0.39 | 0.01 | 0.70 | 0.71 | ✓ |
| D12 | 0.57 | 0.57 | 0.56 | 0.00 | 0.85 | 1.28 | ✓ |
| D13 | 0.58 | 0.67 | 0.58 | 0.00 | 41.84 | 35.61 | ✓ |
| D14 | 0.50 | 0.53 | 0.51 | 0.00 | 3.04 | 1.64 | ✓ |
| D15 | 1.31 | 1.43 | 1.38 | 0.47 | 8.53 | 9.13 | ✓ |
| D16 | 0.36 | 0.37 | 0.37 | 0.00 | 2.63 | 1.48 | ✓ |
| D17 | 0.61 | 0.71 | 0.70 | 0.00 | 12.61 | 13.92 | ✓ |
| D18 | 1.28 | 1.40 | 1.35 | 0.00 | 11.81 | 10.58 | ✓ |
| D19 | 0.55 | 0.75 | 0.69 | 0.01 | 2.03 | 2.27 | ✓ |
| D20 | 0.67 | 0.71 | 0.70 | 0.88 | 0.81 | 0.73 | × |

## H  IMPACT OF FINITE SHOTS (EXTENDED)

The complete results from Section 3.5 are summarized in Table 6. We evaluated the model five times for each shot count in the test sets. We incrementally increased the shot counts until the shot accuracy was within 3% of the analytical accuracy. In Fig. 3, not all experimental results are displayed to avoid overcrowding the figure, which would make it difficult to distinguish individual results.

Table 6: FQN's performance under statistical noise. A-A* represents the difference between the analytical value and the highest accuracy achieved with a finite number of shots.

| Dataset | Number of shots | | | | | | Difference |
|---|---|---|---|---|---|---|---|
| | 10 | 50 | 100 | 500 | 1000 | Analytical (A) | A-A* |
| D1 | 85.5% | 92.9% | 95.6% | – | – | 96.4% | 0.8% |
| D2 | 54.8% | 56.6% | 57.8% | – | – | 59.5% | 1.7% |
| D3 | 60.3% | 84.0% | 86.7% | 98.8% | – | 98.7% | -0.1% |
| D6 | 67.5% | 82.2% | 83.6% | 87.2% | 93.1% | 93.3% | 0.2% |
| D16 | 85.5% | 92.9% | 95.6% | – | – | 97.8% | 2.2% |
| D19 | 44.8% | 45.3% | 50.1% | 58.6% | – | 56.1% | -2.5% |

