# OpenReview forum: "Fully Quanvolutional Networks for Time Series Classification"
_ICLR.cc/2025/Conference — Submitted to ICLR 2025_

### Official Review · Reviewer_kez7 · 2024-10-29

**Soundness:** 3
**Presentation:** 2
**Contribution:** 3
**Rating:** 5
**Confidence:** 3

**Summary:**

The paper introduces a novel Quanvolutional Networks (FQN) for time series classification. The key innovation is the Quanv1D layer, which uses quantum circuits instead of classical convolutional filters to process time series data. The model employs amplitude embedding to encode classical data into quantum states efficiently, requiring fewer qubits than previous approaches. The authors tested FQN on 12 UEA time series datasets and is vaildated against other baseline models.

**Strengths:**

1. The Quanv1D layer design is novel and can handle arbitrary input dimensions, and the paper proves the potential of quantum operations on time series problems

2. The newly design architecture is parameter efficient (2-7x fewer parameters than classical counterparts as the author claimed)

3. Comparable performance against models with much more parameters

**Weaknesses:**

1.	My major concern is experiments. The tasks used in the paper contains 12 datasets out of UEA classification archive which consists of 30 tasks. As it is commonly known that the performance gap among time series models can greatly vary among different datasets, there could be potential dataset set selection bias in this paper. And no explicit explanation of why these specific 12 datasets were chosen.

Besides, the only ‘strong’ baseline is from ModernTCN, which happens to use 10 datasets from UEA 30 (check Appendix A.3 in https://openreview.net/pdf?id=vpJMJerXHU). I’m not commenting on that paper, but unfortunately there’s only 1 dataset (SelfRegulationSCP2) overlapped between the two papers, the other datasets are different.

In this case, as a fair comparison, I suggest the authors to introduce more SOTA time series classification models (Omni-scale cnn, TimesNet), and show the complete experiment results over all 30 UEA datasets or provide a clear explanation of why specific datasets were included/excluded.

2.	The motivation and introduction of quantum operations can be further clarified.

Firstly, for general audience without quantum computing background, it’s better to give some introduction about the basics in the appendix. (i.e. computational basis state in equation 2, definition of shots) along with the benefit of using potential quantum computing.

Secondly, can you clarify the motivation of the proposed method? Do you try to deploy the whole network to quantum computer sometime in the long future or the work is inspired by quantum operations to improve parameter efficiency on classical computer? If the former, I’d like to understand if the other components(activation/matrix operation)of the NN are also quantum operator compatible, and how much it can speed up the algorithm. If the latter, I’d like know by saving the parameter number, how much numerical accuracy does it lose/computational efficiency can it gain?

**Questions:**

1.	By simulating the quantum operation on classical computers, what is the computational overhead of simulating these quantum circuits compared to classical convolutions? And do you have quantitative comparison of computational time to classical time series models?

2.	Can you give a comparison on the numerical performance between the quantum operation vs classical operations in the ideal scenario? For instance, regarding the amplitude embedding, does n qubits lose information or will it behave probabilistically?

---

> ### Author Response · Authors · 2024-11-21
> **Response to Reviewer kez7 (1)**
>
> Questions:
>
> *1. By simulating the quantum operation on classical computers, what is the computational overhead of simulating these quantum circuits compared to classical convolutions? And do you have quantitative comparison of computational time to classical time series models?*
>
> Response: Our study is largely theoretical in nature, where we simulated a quantum environment in a classical computer. For this reason, our proposed layer, Quanv1D, not only performs what a typical convolution layer does but also simulates quantum computations inside it. As such, there would be some computational overheads. However, we coded everything from scratch to fully utilize the batched computational powers of Pytorch. Our simulations were fast and could be completed taking relatively similar time as a regular deep learning model would.
>
> To put the computation difference into perspective, let’s look at the time complexities of both classical convolution and our proposed quanvolution algorithm:
>
> * Classical: $\mathcal{O}(k \times c_{in} \times c_{out})$
> * Quantum: $\mathcal{O}(k^2 \times c_{in}^2 \times c_{out})$
>
> Here, $k$, $c_{in}$, and $c_{out}$ denote the kernel size, number of input channels, and number of output channels/feature maps. We can clearly see how, for quantum computation, the time complexity scales quadratically. The unitary operations inside quantum circuits are matrices with complex numbers, which increases the overhead even further. Nevertheless, with the help of Pytorch and its tensor operations and efficient batched multiplications, we were able to complete the runs in a timely manner. Additionally, we kept the values of $k$ and $c_{in}$ small in general, which helped keep the model manageable. Because the time difference between running other models and our proposed model was not significant (as observed), we did not record the computational time for each epoch or each run.
>
> *2. Can you give a comparison on the numerical performance between the quantum operation vs classical operations in the ideal scenario? For instance, regarding the amplitude embedding, does n qubits lose information or will it behave probabilistically?*
>
> Response: We would like to answer this question from both theoretical and practical perspectives.
>
> In simulations, there won’t be any information loss, as we are directly using the input values as amplitudes of quantum states. However, before the embedding, we normalized the values using a softmax function. As we are altering the input using non-linearity, we can refer to this as exponential scaling rather than direct information loss. However, this introduces two challenges that can cause a difference between convolution and quanvolution. First, in classical convolution, there is a linear relationship (one-to-one mapping) between the filter weights and input. In quanvolution, however, we could not establish this linear mapping due to amplitude embedding. Second of all, the softmax function has its own caveats. For example, it is sensitive to a vector containing a large range of numbers, amplifying larger values disproportionately. This may result in the loss of some finer details or patterns in the data, which could have been crucial during the inference process.
>
> In practical scenarios, there will be direct information loss, mainly pertaining to noisy hardware. For instance, there is a high likelihood of state preparation error when initializing quantum circuits using amplitude embedding. This loss of information will be further intensified by noises related to decoherence, finite shots, etc. Since we lack the hardware to test the information deformity, we cannot provide an exact answer at this moment. However, with our shot-based experiment, we showed that even with statistical noise present, our proposed quanvolution algorithm was efficient.

---

> ### Author Response · Authors · 2024-11-21
> **Response to Reviewer kez7 (2)**
>
> **Weakness 1.**
>
> Response: We acknowledge your concerns regarding the selection bias of test datasets. We should have written the selection criteria more clearly in the paper. Also, we used univariate and multivariate datasets hosted by both UEA [1] and UCR [2] at timeseriesclassification.com. Our writing was not up to par in the first version, which is why such confusions arose. We would like to clarify some of these in this response.
>
> When we first built our model, FQN (built with Quanv1D), we wanted to explore its learning ability when compared to its exact classical counterpart, FCN (built with Conv1d). As a preliminary analysis, we tested FQN and FCN on binary classification datasets and included the only available time series-based quanvolution model, QuanvNet [3], in the mix. While searching for testing data, we found that the archive has 150+ datasets, and among them, around 58 of them were binary classification problems. For our initial analysis, we selected a subset of 12 datasets from these 58. Our selection criteria was primarily based on the domain and its real-life applicability. Additionally, we had to keep some univariate datasets as QuanvNet does not work on multivariate data.
>
> According to Table 1 in the paper, the chosen 12 datasets come from 9 different practical domains. At first, we trained the models on 8 univariate datasets (D1-D8), each belonging to a separate domain. However, while choosing multivariate datasets for our study, we observed that most binary classification problems with multivariate features (in the archive) originated from the EEG domain. As such, we randomly selected 4 EEG multivariate datasets from the repository. The decision to retain all 4 of the EEG datasets was motivated by the practical applicability of the domain along with the challenging nature of the datasets that originates from high variability and low statistical power [4-6].
>
> We only decided to test FQN's performance against the SOTA time series classifier, ModernTCN [7], after compiling the initial results and noticing that FQN was performing slightly better than FCN on average. When we checked the ModernTCN paper and saw that they have used different datasets for classification (except one overlap, SelfRegulationSCP2), we considered adding these datasets for our study. But because the authors of ModernTCN did not claim their datasets were the benchmarks, we opted not to proceed in that direction at that time. Flowformer [8] was the first to introduce these ten datasets for time series classification, which inspired ModernTCN. However, these ten datasets were never considered as benchmarks in the literature, nor was their usage justified beyond data diversity.
>
> [1] Bagnall, Anthony, et al. "The UEA multivariate time series classification archive, 2018." arXiv preprint arXiv:1811.00075 (2018).
>
> [2] Dau, Hoang Anh, et al. "The UCR time series archive." IEEE/CAA Journal of Automatica Sinica 6.6 (2019): 1293-1305.
>
> [3] Rivera-Ruiz, Mayra Alejandra, et al. "1d quantum convolutional neural network for time series forecasting and classification." Mexican International Conference on Artificial Intelligence. Cham: Springer Nature Switzerland, 2023.
>
> [4] Davoudi, Saeideh, et al. "Inter-individual variability during neurodevelopment: an investigation of linear and nonlinear resting-state EEG features in an age-homogenous group of infants." Cerebral Cortex 33.13 (2023): 8734-8747.
>
> [5] Larson, Michael J., and Kaylie A. Carbine. "Sample size calculations in human electrophysiology (EEG and ERP) studies: A systematic review and recommendations for increased rigor." International Journal of Psychophysiology 111 (2017): 33-41.
>
> [6] Button, Katherine S., et al. "Power failure: why small sample size undermines the reliability of neuroscience." Nature reviews neuroscience 14.5 (2013): 365-376.
>
> [7] Luo, Donghao, and Xue Wang. "Moderntcn: A modern pure convolution structure for general time series analysis." The Twelfth International Conference on Learning Representations. 2024.
>
> [8] Wu, Haixu, et al. "Flowformer: Linearizing transformers with conservation flows." arXiv preprint arXiv:2202.06258 (2022).

---

> > ### Author Response · Authors · 2024-11-21
> > **Response to Reviewer kez7 (2) Continued**
> >
> > However, we strongly agree with your assessment that we should have selected and tested all the datasets for a more robust comparison, and the archive's authors have also expressed similar views [1]. However, due to time and resource constraints, we could not run experiments on every dataset available earlier, a limitation that we want to address in the future. For now, we have conducted some additional experiments on multiclass problems to make our analysis more comprehensive. Here are the results we have obtained so far:
> >
> > |                       | Test accuracy |         |           |              | Parameter ratio |        |           |              |
> > |-----------------------|---------------|---------|-----------|--------------|-----------------|--------|-----------|--------------|
> > |        Dataset        |      FQN      |   FCN   | ModernTCN |   QuanvNet   |       FQN       |   FCN  | ModernTCN |   QuanvNet   |
> > |  EthanolConcentration |     29.14%    |  28.57% |   29.71%  |       -      |    x1 (4858)    |  x6.61 |  x107.34  |       -      |
> > |      NerveDamage      |     99.51%    |  99.51% |   59.02%  |    98.54%    |    x1 (3440)    |  x3.42 |   x21.88  |    x27.02    |
> > |  DiatomSizeReduction  |     98.13%    | 100.00% |   99.69%  |    89.06%    |    x1 (6098)    |  x6.37 |   x8.65   |    x15.32    |
> > | HandMovementDirection |     34.47%    |  31.06% |   36.60%  |       -      |    x1 (1488)    |  x2.59 |   x96.82  |       -      |
> > |    SyntheticControl   |     98.66%    |  98.83% |   96.83%  |    96.00%    |    x1 (2988)    | x13.02 |   x11.12  |    x31.62    |
> > |        Average        |     71.98%    |  71.60% |   64.37%  | _Incomplete_ |        x1       |  x6.40 |   x49.16  | _Incomplete_ |
> >
> >
> > We aim to add some more experiments and compile them for the final submission. Also, the results for QuanvNet are incomplete because it cannot handle multivariate datasets.
> >
> > On the topic of baselines, our aim was never to create a new time series classifier that would surpass all the existing models. Although there are other time series classifiers in the literature, since the ModernTCN paper reported it to be outperforming other classifiers significantly, at that moment, we decided not to include other models and thought to be complete in terms of baselines. As of now, we are focused on our own model in terms of quantum-aware optimization, custom activation functions, better performance, robustness against noise, etc. Also, we could not test our model on other data learning tasks like forecasting and anomaly detection, which would require further refinements in the model design. We would definitely include other baselines and, of course, more datasets when we are able to achieve the aforesaid goals.
> >
> > [1] Dau, Hoang Anh, et al. "The UCR time series archive." IEEE/CAA Journal of Automatica Sinica 6.6 (2019): 1293-1305.

---

> ### Author Response · Authors · 2024-11-21
> **Response to Reviewer kez7 (3)**
>
> **Weakness 2.**
>
> Response:
>
> We appreciate your suggestion about introducing the basics of quantum in the appendix. We are currently incorporating it in the manuscript.
>
> For your second point, our initial motivation for building this model was not necessarily to improve parameter efficiency, but rather to create a model that could potentially be implemented in real quantum computers. However, we should point out that the whole network cannot be implemented on a quantum computer. For instance, we can implement the quanvolution part as it follows all the basics of quantum mechanics, but for the activation layers, the computation has to be done in traditional computers. This is due to the inability of current quantum computers to incorporate nonlinearities. Nevertheless, since the activation part is very lightweight, $\mathcal{O}(n)$, we will still be able to gain the computational advantage provided by a quantum computer for the whole model. As to how much it can speed up the algorithm, we cannot comment on that yet as we do not have access to quantum hardware capable of handling this level of implementation. But, after exploring existing literature on this topic on how to implement such models in real life, we came across two studies that built an artificial neural network for quantum processing units but applied the nonlinearities in a classical computer, thereby following a hybrid training scheme and still getting the benefits from the quantum computation [1, 2]. We believe our approach can be of similar stature.
>
> [1] Tacchino, Francesco, et al. "An artificial neuron implemented on an actual quantum processor." npj Quantum Information 5.1 (2019): 26.
>
> [2] Tacchino, Francesco, et al. "Quantum implementation of an artificial feed-forward neural network." Quantum Science and Technology 5.4 (2020): 044010.

---

> > ### Comment · Reviewer_kez7 · 2024-11-25
> > **reply to author rebuttal**
> >
> > I thank the authors for answering my questions, I also checked the comments from other reviewers. I strongly agree with Reviewer fVyg, that a full dataset should be used for proper evaluation.
> >
> > Regarding my second point on weakness, I thank the authors for clarifying the initial motivation of the paper. As the authors pointed out the model "was not necessarily to improve parameter efficiency" while still replying on hybrid approach to work, it is difficult to justfy the practical contribution at this stage for both sides (classical computer or quantum computer).
> >
> > I would keep my score for now.

---

> > > ### Author Response · Authors · 2024-11-30
> > > **Response to "reply to author rebuttal" by Reviewer kez7**
> > >
> > > Thank you for your suggestion on incorporating more datasets. We will definitely look into it in future submissions.
> > >
> > > We are currently in the NISQ era, and on our way to more advanced fault-tolerant quantum computers with increased number of qubits. At this current stage, obviously, this level of implementation (FQN) cannot be performed on real hardware. As such, we tried to showcase some advantages of FQN even if it's used in classical computers (with simulators). Our overall performance indicate that FQN can potentially be used for instances where classical models overfit and thus cannot generalize. And when quantum computers would eventually be more accessible, to verify whether FQN can be used or not (in real life), we trained FQN subjected to statistical noise and got positive results.
> > >
> > > We have updated our *Concluding Remarks* section to better clarify where FQN can be useful and how it can be implemented in the near future. We would love to hear your insights and suggestions on the revised manuscript, as they will help us identify areas for improvement.

---

### Official Review · Reviewer_fVyg · 2024-10-31

**Soundness:** 2
**Presentation:** 2
**Contribution:** 2
**Rating:** 3
**Confidence:** 3

**Summary:**

In this paper, a new method built upon the quanvolutional operator, which is derived by **quantum circuits in place of standard convolutional filters**, is proposed. To overcome the exclusive reliance of quanvolutional neural networks on classical layers, as well as their limited application to multi-dimensional data with potential scalability issues, authors propose a novel **Quanv1D layer** for time series data, which acts as the main building block in a lightweight architecture, the so-called Fully Quanvolutional Network (FQN). For increased efficiency, they also propose the incorporation of *amplitude embedding for data encoding* that enables minimal qubit usage. The proposed Quanv1D layers are stacked with *increased dilation rates* to enable larger receptive fields, followed by a final projection layer. Authors evaluate FQN architecture in time series classification, achieving competitive performance with standard CNN-based architectures and other quanvolutional-based architectures while remaining significantly more efficient in terms of memory.

**Strengths:**

Important strong aspects of the proposed method and study are the following:
1. Authors tackle the very promising and *relatively underexplored* methodological field of quantum machine learning.
2. They propose a building block that combines ideas from *1D convolutional layers and quantum circuits* (Quanv1D layer) and design it specifically for **multi-dimensional time series** data.
3. Importantly, the proposed Quanv1D layer adopts the *standard hyperparameters of 1D convolution*, such as kernel size, stride, dilation, and padding, enabling their straightforward application.
4. Their proposed architecture, built upon an efficient amplitude embedding layer and stacked Quanv1D layers, remains *light in terms of parameters while achieving competitive performance* in few time series classification datasets. Additionally, while relying solely on quantum layers, it showcases their promising application, which is not affected by other standard neural networks.

**Weaknesses:**

Significant weaknesses of the presented work are summarized as follows:
1. **Presentation of Related Works:** Related works in quantum-based architectures are not thoroughly presented as preliminaries for the proposed method. It is not clear if the main design of the proposed quanvolutional layer significantly expands previous designs beyond the incorporation of user-friendly hyperparameters. This raises doubts about the novelty of the proposed method, making it mostly limited to the selection of the embedding layer.
2. **Experimental Design:** The experimental setup differs from common baselines in the time series classification field. Specifically, the 12 datasets from UEA used by the authors are totally different from the ones used in recent studies [1,2] and commonly used benchmarks [3]. Additionally, the comparisons are limited to two CNN-based and one quant-based network, excluding several common baselines in the field. These choices raise questions about the generalization performance of the proposed method beyond the selected datasets/baselines.
3. **Applicability to Real Hardware and Impact:** The performance achieved by the proposed FQN model in time series classification is, in several cases, inferior to CNN-based baselines. Yet, the number of parameters used by FQN is significantly lower, which makes it computationally attractive and promising for large-scale applications. On the other hand, as mentioned by the authors, the proposed method remains a theoretical framework tested on conventional computers, which questions the potential impact of this contribution if quantum computations are enabled. This is further confirmed by the presented simulation of FQN on finite shots, where the performance was not matched for most datasets to the one achieved with analytical expectation values.

[1] Luo, D., & Wang, X. (2024). Moderntcn: A modern pure convolution structure for general time series analysis. In The Twelfth International Conference on Learning Representations.

[2] Wu, H., Hu, T., Liu, Y., Zhou, H., Wang, J., & Long, M. (2022). Timesnet: Temporal 2d-variation modeling for general time series analysis. arXiv preprint arXiv:2210.02186.

[3] https://github.com/thuml/Time-Series-Library

**Questions:**

- **Q1 - Datasets:** Based on weakness (2), could the authors explain the selection of this subset of UEA, which differs from the standard 10 preferred datasets in most studies? Have you conducted experiments on the whole data repository? Similarly, most studies in time series classification first evaluate methods on the univariate UCR archive [1].
- **Q2 - Experimental Evaluation:** Incorporating additional baselines or tasks could further support performance comparisons in favor of your proposed architecture. Please justify your choices if this is not possible.
- **Q3 - Related Works as Preliminaries:** To better position your contribution in the field of quantum machine learning, it would be essential to clarify better where the technical design of Quanv1D differs from previous quanvolutional layers (weakness (1)). For instance, you can use relevant notations from the literature as a “preliminaries” section in the method.

---

> ### Author Response · Authors · 2024-11-20
> **Response to Reviewer fVyg (1)**
>
> Questions:
> *1. Based on weakness (2), could the authors explain the selection of this subset of UEA, which differs from the standard 10 preferred datasets in most studies? Have you conducted experiments on the whole data repository? Similarly, most studies in time series classification first evaluate methods on the univariate UCR archive [1].*
>
> Response: When we first built our model, FQN (built with Quanv1D), we wanted to explore its learning ability when compared to its exact classical counterpart, FCN (built with Conv1d). As a preliminary analysis, we tested FQN and FCN on binary classification datasets and included the only available time series-based quanvolution model, QuanvNet [1], in the mix. While searching for testing data, we found that the archive [2-4] has 150+ datasets, and among them, around 58 of them were binary classification problems. For our initial analysis, we selected a subset of 12 datasets from these 58. Our selection criteria was primarily based on the domain and its real-life applicability. Additionally, we had to keep some univariate datasets as QuanvNet does not work on multivariate data.
>
> According to Table 1, the chosen 12 datasets come from 9 different practical domains. At first, we trained the models on 8 univariate datasets (D1-D8), each belonging to a separate domain. However, while choosing multivariate datasets for our study, we observed that most binary classification problems with multivariate features (in the archive) originated from the EEG domain. As such, we randomly selected 4 EEG multivariate datasets from the repository. The decision to retain all 4 of the EEG datasets was motivated by the practical applicability of the domain along with the challenging nature of the datasets that originates from high variability and low statistical power [5-7].
>
> We only decided to test FQN's performance against the SOTA time series classifier, ModernTCN [8], after compiling the initial results and noticing that FQN was performing slightly better than FCN on average. When we checked the ModernTCN paper and saw that they have used different datasets for classification (except one overlap, SelfRegulationSCP2), we considered adding these datasets for our study. But because the authors of ModernTCN did not claim their datasets were the benchmarks, we opted not to proceed in that direction at that time. Flowformer [9] was the first to introduce these ten datasets for time series classification, which inspired TimesNet [10] and later, ModernTCN. However, these ten datasets were never considered as benchmarks in the literature, nor was their usage justified beyond data diversity. The TSLib library built their classification section following the Flowformer repository as acknowledged in their repo [11]. As such, they also suggested these ten datasets but never claimed them to be benchmarks. This is the reason we didn't include any additional datasets in our initial testing.
>
> [1] Henderson, Maxwell, et al. "Quanvolutional neural networks: powering image recognition with quantum circuits." Quantum Machine Intelligence 2.1 (2020): 2.
>
> [2] Dau, Hoang Anh, et al. "The UCR time series archive." IEEE/CAA Journal of Automatica Sinica 6.6 (2019): 1293-1305.
>
> [3] Bagnall, Anthony, et al. "The UEA multivariate time series classification archive, 2018." arXiv preprint arXiv:1811.00075 (2018).
>
> [4] https://www.timeseriesclassification.com/dataset.php
>
> [5] Davoudi, Saeideh, et al. "Inter-individual variability during neurodevelopment: an investigation of linear and nonlinear resting-state EEG features in an age-homogenous group of infants." Cerebral Cortex 33.13 (2023): 8734-8747.
>
> [6] Larson, Michael J., and Kaylie A. Carbine. "Sample size calculations in human electrophysiology (EEG and ERP) studies: A systematic review and recommendations for increased rigor." International Journal of Psychophysiology 111 (2017): 33-41.
>
> [7] Button, Katherine S., et al. "Power failure: why small sample size undermines the reliability of neuroscience." Nature reviews neuroscience 14.5 (2013): 365-376.
>
> [8] Luo, Donghao, and Xue Wang. "Moderntcn: A modern pure convolution structure for general time series analysis." The Twelfth International Conference on Learning Representations. 2024.
>
> [9] Wu, Haixu, et al. "Flowformer: Linearizing transformers with conservation flows." arXiv preprint arXiv:2202.06258 (2022).
>
> [10] Wu, Haixu, et al. "Timesnet: Temporal 2d-variation modeling for general time series analysis." arXiv preprint arXiv:2210.02186 (2022).
>
> [11] https://github.com/thuml/Time-Series-Library?tab=readme-ov-file#acknowledgement

---

> > ### Author Response · Authors · 2024-11-20
> > **Response to Reviewer fVyg (1) continued**
> >
> > For the first version of our manuscript, we only conducted experiments on 12 binary classification datasets. However, currently we are conducting experiments on multi-class problems following the same data selection methodology and will include the results in the updated manuscript. The results of the multiclass classification so far:
> >
> > |                       | Test accuracy |         |           |              | Parameter ratio |        |           |              |
> > |-----------------------|---------------|---------|-----------|--------------|-----------------|--------|-----------|--------------|
> > |        Dataset        |      FQN      |   FCN   | ModernTCN |   QuanvNet   |       FQN       |   FCN  | ModernTCN |   QuanvNet   |
> > |  EthanolConcentration |     29.14%    |  28.57% |   29.71%  |       -      |    x1 (4858)    |  x6.61 |  x107.34  |       -      |
> > |      NerveDamage      |     99.51%    |  99.51% |   59.02%  |    98.54%    |    x1 (3440)    |  x3.42 |   x21.88  |    x27.02    |
> > |  DiatomSizeReduction  |     98.13%    | 100.00% |   99.69%  |    89.06%    |    x1 (6098)    |  x6.37 |   x8.65   |    x15.32    |
> > | HandMovementDirection |     34.47%    |  31.06% |   36.60%  |       -      |    x1 (1488)    |  x2.59 |   x96.82  |       -      |
> > |    SyntheticControl   |     98.66%    |  98.83% |   96.83%  |    96.00%    |    x1 (2988)    | x13.02 |   x11.12  |    x31.62    |
> > |        Average        |     71.98%    |  71.60% |   64.37%  | _Incomplete_ |        x1       |  x6.40 |   x49.16  | _Incomplete_ |
> >
> > The results for QuanvNet are incomplete because it cannot handle multivariate datasets.

---

> ### Author Response · Authors · 2024-11-20
> **Response to Reviewer fVyg (2)**
>
> *2. Incorporating additional baselines or tasks could further support performance comparisons in favor of your proposed architecture. Please justify your choices if this is not possible.*
>
> Response: Our aim was never to create a new time series classifier that would surpass all the existing models. Our goal was only to create a quantum-equivalent layer for Conv1d. As Quanv1D is a direct equivalent of Conv1d in the quantum domain, our main baseline for testing FQN (built with only Quanv1D) was its classical counterpart, FCN (built with only Conv1d). To test the efficacy of FQN with existing quantum models, we included QuanvNet in the comparison because it is the only existing quanvolutional model that works directly with time series data. In addition to these two models, we brought in the convolution-based ModernTCN, the SOTA time series classifier, to provide a comprehensive study of FQN’s performance. Although there are other time series classifiers in the literature (like TimesNet, Crossformer, Flowformer, etc.), since ModernTCN is currently the best and outperformed all other models substantially, we decided not to include other models. In light of this, we believe our study to be complete in terms of baselines.
>
> However, in terms of tasks, we still have a long way to go. Our FQN, as of now, is capable of classification only. But we haven’t explored forecasting or imputation tasks yet. Since we would like to hold on to some of the advantages of our model (like parameter efficiency and self-regularization) for these tasks, we have to keep it fully quanvolutional. However, Quanv1D can only give output between -1 and +1, which can potentially hinder data learning. Hence, we are currently working on developing activation functions to expand the output range for regression and imputation tasks. In addition, we believe there is still room for improvements in the classification model itself (performance, hyperparameters, optimization, non-linearity, etc.). At this moment, we will not explore other data learning tasks, but definitely will in the future.

---

> ### Author Response · Authors · 2024-11-20
> **Response to Reviewer fVyg (3)**
>
> *3. To better position your contribution in the field of quantum machine learning, it would be essential to clarify better where the technical design of Quanv1D differs from previous quanvolutional layers (weakness (1)). For instance, you can use relevant notations from the literature as a “preliminaries” section in the method.*
>
> Response: Our work is an extension of the existing quanvolutional methods, drawing inspiration from the classical convolution. We designed our own Quanv1D layer, which works similar to Conv1d but has significantly fewer parameters. Additionally, our Quanv1d has a self-regularizing property, which makes it perform better on harder-to-generalize datasets when compared to classical convolution. On top of that, we addressed an earlier limitation of quanvolutional models, where only the first layer was quantum-based and all the subsequent layers were classical, and developed a fully quanvolutional model (i.e., all the processing layers of the model are Quanv1D). However, we acknowledge that our writing for the initial version is a bit lacking and might not have given the correct impression. We are currently updating the manuscript, taking all the reviews into account. We hope to present the updated manuscript within a few days.
>
> We appreciate your suggestion of including a "preliminaries" section where we could showcase how our proposed layer differs from the previous versions, and we will definitely include it.

---

> ### Author Response · Authors · 2024-11-20
> **Response to Reviewer fVyg (4)**
>
> We would also like to address weakness 3.
>
> In Section 3.4, we included a description for the behavior of different datasets under finite shots. Since the finite shots are just average values, by the law of large numbers, the performance of the model on all the datasets should reach the theoretical value if enough shots are used [1]. In Figure 3, D5 (PowerCons) does not manage to reach theoretical accuracy within 500 shots, and we acknowledge this limitation. As such, we managed a better GPU and decided to re-run D5 with 1000 shots. Our experiments show that shot-based FQN nearly reaches (within 3%) the analytical accuracy for D5, a huge improvement from the previous deviation (around 10%). We have also included one more dataset in our shot-based learning, Blink. Here is the current performance list for shot-based learning:
>
> | Dataset            | Analytical accuracy | Shot-based accuracy | Deviation | Required shots |
> |--------------------|---------------------|---------------------|-----------|----------------|
> | Chinatown          | 97.5%               | 96.71%              | -0.79%    | 500            |
> | FingerMovements    | 59.50%              | 58.56%              | -0.94%    | 500            |
> | SharePriceIncrease | 58.1%               | 59.22%              | 1.12%     | 500            |
> | PowerCons          | 95.80%              | 93.06%              | -2.74%    | 1000           |
> | Blink              | 99.00%              | 97.47%              | -1.53%    | 500            |
> | Average            | 82.0%               | 81.00%              | -0.98%    | N/A            |
>
> In the updated manuscript, we plan to present at least one or two multiclass classification datasets under shot-based learning.
>
> [1] Evans, Michael J., and Jeffrey S. Rosenthal. Probability and statistics: The science of uncertainty. Macmillan, 2004.

---

> > ### Comment · Reviewer_fVyg · 2024-11-25
> > **Reply to Rebuttal**
> >
> > I would like to thank the authors for their efforts in addressing my concerns. I appreciate their time and willingness to improve their work and clarify blurry points.
> >
> > Although I find potential in the proposed quantum-based convolutional layer, tested on time series data, I still have **several doubts about raising my scores**:
> > - **Datasets and Baselines:** I really appreciate the effort of the authors to incorporate more datasets from the UCR and UEA archives; however, from what I understand, there is still a long way to go before the results are complete. With respect to the baselines, I respectfully disagree with the authors. In time series classification, the community chooses to include a long list of baseline architectures, including deep learning and machine learning ones, since even relatively simple methods show sota performances (please refer to ROCKET, InceptionTime). In this sense, unfortunately, I think the experimental evaluation of FQN is incomplete, at least for the relatively well-explored task of time series classification.
> > - **Positioning of the paper:** As far as I understand, you position the paper across other quantum-based DL methods, in the fact that you proposed a quantum-based equivalent of 1d CNN (similar parameters, etc.). Still, a 'preliminaries' section could provide a clearer positioning and introduction in the proposed FQN for non-experts in the field, and I am satisfied that you agree with that. Additionally, there is a conceptual gap between proposing a quantum-based 1d CNN and applying this to the particular case of time series. Time series data encompass challenging properties, such as autocorrelation, periodicities, and multivariate dependencies, which necessitate careful modeling, and it is not clear to me how FQN could contribute to time series modeling in these directions. It would be more straightforward to validate a quantum-based equivalent of 1d CNN to standard discrete sequence modeling before extending it to time series.
> > - **Practical applicability of FQN:** Concerning the behavior of different datasets under finite shots, it is understandable that enough shots are required to match the theoretical performance values. The updated results on the presented simulation of FQN on finite shots support the applicability of the proposed method slightly better yet remain limited to a few selected datasets for non-specified reasons. Weakness 3 in my review, about hardware and applicability, also supports my doubts.
> >
> > For all the above reasons, I prefer to keep my score.

---

> > > ### Author Response · Authors · 2024-11-30
> > > **Response to "Reply to Rebuttal" by Reviewer fVyg**
> > >
> > > 1. We would like to clarify that our goal was not to present a new SOTA in time series classification or to establish FQN as the new industry standard. Instead, our goal was to build a functional 1D quanvolution model that performs similarly to 1D convolution. Since 1D convolution is generally applied to temporal data or signals, we compared these two approaches within the context of time series classification. Once again, we did not design Quanv1D with time series in mind; rather, our goal was to create a quantum equivalent of Conv1D, and we selected time series as a preliminary testing domain. We agree with your observation that there is still much room for improvement, and we plan to incorporate additional datasets as we continue to refine our algorithm.
> > >
> > > 2. Thank you for suggesting that we apply our model to discrete sequence data. We are certainly interested in exploring this domain in the future. However, for this study, as time series is a relatively well-explored domain (as you mentioned), we aimed to determine whether 1D quanvolution performs and behaves similarly to 1D convolution in such a domain. This allowed us to derive interpretable insights, which, otherwise, would have been difficult in a less explored field.
> > >
> > > 3. We randomly selected 6 datasets out of the 20 for this section. Unfortunately, we could not evaluate our approach on all 20 datasets due to time and computational constraints. However, we will try to include all datasets in our future experiments.

---

### Official Review · Reviewer_YA86 · 2024-11-02

**Soundness:** 2
**Presentation:** 1
**Contribution:** 2
**Rating:** 3
**Confidence:** 3

**Summary:**

This paper proposes a novel layer based on quantum computation, and defines a model for time series classification that, on average, surpass previous models. Overall, the idea may sound but the paper requires a refactoring in terms of writing, style, depth of investigation and discussion, and motivation.

**Strengths:**

1) The proposed method achieves interesting results also with respect to ModernTCN.
2) The motivation and the advantages with respect to QuanvNet are clear.

**Weaknesses:**

1) Writing needs huge improvements. The authors begin both the abstract and the introduction talking about quanvolutional stuff without explaining what they are or the reason to use them. Without a proper introduction, it is arduous to understand the novelty of this paper. Overall, the paper seems a coding report.
2) Equation (1) does not have any non-quantum comparison so it is difficult to understand the differences.
3) Experiments focus on time series classification, while the method needs a broader discussion on the implications and possible applications.
4) I guess more references exist together with QuanvNet, and a deeper investigation is required.

Minor comments:

The Saxon genitive should be avoided in scientific writing, although I know that both ChatGPT and Grammarly suggest using it. However, I suggest the authors remove all the Saxon genitives from the paper.

**Questions:**

1) As the authors correctly point out in 3.5, the proposed method avoids overfitting while the counterpart FCN overfits. Together with the demonstration that FQN naturally acts as a regularization technique, some experiments should be performed: it would important to prove it by defining the FCN model with the same number of paramaters of FQN to show that the overfitting is not due to the higher number of params.

---

> ### Author Response · Authors · 2024-11-20
> **Response to Reviewer YA86 (1)**
>
> Question:
> *As the authors correctly point out in 3.5, the proposed method avoids overfitting while the counterpart FCN overfits. Together with the demonstration that FQN naturally acts as a regularization technique, some experiments should be performed: it would important to prove it by defining the FCN model with the same number of paramaters of FQN to show that the overfitting is not due to the higher number of params.*
>
> Response: This concern led us to rerun FCN with the same parameters as FQN. To ensure equal parameters, we have utilized grid searching to determine the hyperparameters. The loss values from the runs are listed below:
>
> | Dataset | FQN   |      |      | FCN   |      |      | Param count |      |
> |---------|-------|------|------|-------|------|------|-------------|------|
> |         | Train | Val  | Test | Train | Val  | Test | FQN         | FCN  |
> | D1      | 0.27  | 0.29 | 0.31 | 0.00  | 0.01 | 0.17 | 856         | 856  |
> | D2      | 0.61  | 0.68 | 0.68 | 0.03  | 3.21 | 2.96 | 908         | 908  |
> | D3      | 0.55  | 0.59 | 0.57 | 0.04  | 0.64 | 0.72 | 2576        | 2576 |
> | D4      | 0.43  | 0.60 | 0.55 | 0.01  | 1.09 | 0.59 | 1808        | 1808 |
> | D5      | 0.40  | 0.44 | 0.43 | 0.00  | 0.02 | 0.02 | 4434        | 4434 |
> | D6      | 0.38  | 0.39 | 0.41 | 0.13  | 0.48 | 0.27 | 3200        | 3200 |
> | D7      | 0.48  | 0.82 | 0.57 | 0.00  | 3.10 | 1.38 | 3400        | 3401 |
> | D8      | 0.58  | 0.67 | 0.58 | 0.55  | 0.72 | 0.57 | 1166        | 1166 |
> | D9      | 0.36  | 0.37 | 0.37 | 0.00  | 0.00 | 0.01 | 3618        | 3618 |
> | D10     | 0.61  | 0.71 | 0.70 | 0.64  | 0.75 | 0.71 | 4882        | 4882 |
> | D11     | 0.55  | 0.75 | 0.69 | 0.61  | 0.85 | 0.75 | 728         | 728  |
> | D12     | 0.67  | 0.71 | 0.70 | 0.68  | 0.70 | 0.71 | 524         | 525  |
>
> We couldn’t manage to make the parameters exactly equal for D7 and D12. Now, this setup has improved overfitting in the case of D6, D8, D10-D12. But overfitting is still present for D1-D4 and D7. Improving overfitting also changed the **test accuracies**, as shown below:
>
> | Dataset | FQN   | FCN (same param) | FCN (original) | Change |
> |---------|-------|------------------|----------------|--------|
> | D1      | 97.5% | 97.0%            | 96.4%          | 0.5%   |
> | D2      | 58.1% | 59.5%            | 59.0%          | 0.5%   |
> | D3      | 75.3% | 78.6%            | 80.9%          | -2.3%  |
> | D4      | 77.0% | 81.0%            | 74.5%          | 6.5%   |
> | D5      | 95.8% | 99.2%            | 98.9%          | 0.3%   |
> | D6      | 93.9% | 91.5%            | 90.3%          | 1.2%   |
> | D7      | 79.4% | 73.9%            | 73.9%          | 0.0%   |
> | D8      | 72.9% | 71.0%            | 74.1%          | -3.2%  |
> | D9      | 99.0% | 99.6%            | 99.9%          | -0.3%  |
> | D10     | 53.7% | 52.4%            | 44.7%          | 7.6%   |
> | D11     | 59.5% | 52.8%            | 55.7%          | -2.9%  |
> | D12     | 53.2% | 47.4%            | 51.1%          | -3.7%  |
> | Average | 76.3% | 75.3%            | 75.0%          | 0.4%   |
>
> A decrease in overfitting led to better performance in some cases, as observed from the table. However, in some cases, the performance degraded. The accuracy improvement verifies our initial hypothesis of performance degradation for overfitting in FCN. Additionally, this analysis verifies our theoretical finding that FQN effectively combats overfitting because of the chosen ansatz and its self-regularization ability, not just because it has fewer parameters.

---

> ### Author Response · Authors · 2024-11-20
> **Response to Reviewer YA86 (2)**
>
> Response to weaknesses 1 and 3: We acknowledge the fact that the writing is not up to par. We rushed a bit before submission. We are currently updating the manuscript, taking into account all the reviews. We should be able to upload the updated manuscript within a few days.
>
> Response to weakness 2: Equation (1) in the paper refers to the change in sequence length after a convolution operation [1]. It is a non-quantum equation, one on which we built our quanvolutional layer. We tried to make our proposed layer as close to non-quantum convolution as possible. As such, both our quanvolutional layer and the classical convolutional layer follow the same equation. Currently, our writing in that section is confusing, so we will update the manuscript to clarify.
>
> Response to weakness 4: We conducted a systematic search using four databases (Scopus, Web of Science, IEEE Xplore, and ACM Digital Library) for collecting relevant references. In the literature, there are two variants of quantum convolution: kernel-based quantum convolution (also called quanvolution), which is based on quantum circuits replacing the traditional convolutional filters [2], and circuit-based quantum convolution, which takes a CNN model and turns it into a quantum circuit [3]. Since our FQN falls under the first type of quantum convolution, we did not consider models from the second type for analysis.
>
> Most layer-based works on time series data use image-based quanvolution to transform the data (two-dimensional input) into scalograms or spectrograms [4–8]. Through our systematic search, we found only one study that directly used a quanvolution layer-based model for time series analysis—QuanvNet [9], which we have included in our study. While updating the paper, we intend to include this information in detail along with the systematic search results provided as supplementary materials.
>
>
> [1] https://pytorch.org/docs/stable/generated/torch.nn.Conv1d.html
>
> [2] Henderson, Maxwell, et al. "Quanvolutional neural networks: powering image recognition with quantum circuits." Quantum Machine Intelligence 2.1 (2020): 2.
>
> [3] Cong, Iris, Soonwon Choi, and Mikhail D. Lukin. "Quantum convolutional neural networks." Nature Physics 15.12 (2019): 1273-1278.
>
> [4] Li, Yue, et al. "Detection and Identification of Power Quality Disturbance Signals in New Power System Based on Quantum Classic Hybrid Convolutional Neural Networks." International Conference on Data Security and Privacy Protection. Singapore: Springer Nature Singapore, 2024.
>
> [5] Savla, Aansh, et al. "GQNN: Greedy Quanvolutional Neural Network Model." International Conference on Image Processing and Capsule Networks. Cham: Springer International Publishing, 2022.
>
> [6] Sridevi, S., et al. "Quanvolution neural network to recognize arrhythmia from 2D scaleogram features of ECG signals." 2022 International Conference on Innovative Trends in Information Technology (ICITIIT). IEEE, 2022.
>
> [7] Prabhu, Sharanya, et al. "QuCardio: Application of Quantum Machine Learning for Detection of Cardiovascular Diseases." IEEE Access 11 (2023): 136122-136135.
>
> [8] Yang, Chao-Han Huck, et al. "Decentralizing feature extraction with quantum convolutional neural network for automatic speech recognition." ICASSP 2021-2021 IEEE International Conference on Acoustics, Speech and Signal Processing (ICASSP). IEEE, 2021.
>
> [9] Rivera-Ruiz, Mayra Alejandra, et al. "1d quantum convolutional neural network for time series forecasting and classification." Mexican International Conference on Artificial Intelligence. Cham: Springer Nature Switzerland, 2023.

---

> > ### Comment · Reviewer_YA86 · 2024-11-30
> > **Response to rebuttal**
> >
> > I would like to thank the authors for their reply and their clarifications.
> > From the tables above, I am still convinced that more investigations should be performed on overfitting and on the comparison among the methods with fair and equal number of parameters.
> >
> > Also, I was waiting for the improved version of the paper in terms of writing but the authors have not uploaded it yet.
> >
> > Overall, I think that this paper may be valuable but more investigations and much more details should be included, in terms of experiments and writing, therefore I will keep my score.

---

> > > ### Author Response · Authors · 2024-11-30
> > > **Response to "Response to rebuttal" by Reviewer YA86**
> > >
> > > We uploaded the revised manuscript at approximately 3:30 PM UTC on November 27, well before our submission deadline (November 27 End-of-day AoE). Additionally, we provided a detailed revision summary around 6:20 AM UTC, November 28 for your review. We sincerely apologize if there were any notification issues from the venue.
> > >
> > > Regarding your suggestion for a "same parameter" comparison, we conducted the experiment across all 20 datasets. The results (Section 3.4, Table 2 of the updated manuscript) indicate that FQN still outperforms FCN in terms of regularization. This improvement is primarily attributed to FQN's self-regularization properties, as discussed in Section 3.4. These findings suggest that the advantage of FQN does not stem solely from its lower parameter count but rather from its self-regularization capabilities.

---

### Official Review · Reviewer_Zjmd · 2024-11-02

**Soundness:** 2
**Presentation:** 2
**Contribution:** 2
**Rating:** 6
**Confidence:** 2

**Summary:**

## Summary of the Paper's Contribution
The paper introduces a novel quantum convolutional layer, Quanv1D, and a fully quantum convolutional network (Fully Quanvolutional Networks, FQN), for time series classification tasks. By using amplitude embedding, the method reduces qubit requirements and demonstrates competitive experimental results on 12 time series classification datasets. This work highlights the potential of quantum models in time series classification.

**Strengths:**

## Strengths:
1.	**Innovation**: The Quanv1D layer and FQN architecture broaden the applicability of quantum convolutional networks, offering a fresh perspective for combining quantum computing and time series analysis.
2.	**Comprehensive Experimental Design**: The experiments cover multiple time series classification datasets, demonstrating FQN’s comparable or superior performance to current state-of-the-art models on various datasets.
3.	**Transparent Limitations Analysis**: The paper thoroughly discusses the gap between the model’s theoretical framework and hardware implementation, especially addressing the quantum hardware limitations in the NISQ era.

**Weaknesses:**

## Weaknesses:
1.	**Limited Real Hardware Applicability**: As the experiments were all conducted in simulation, there is currently a lack of data on FQN’s performance on real quantum hardware, which limits its feasibility for practical applications.
2.	**Limited Task Scope**: FQN has been validated only on classification tasks, with no exploration of applications like time series forecasting or imputation.
3.	**Resource Demands Remain High**: Although amplitude embedding reduces qubit requirements, circuit depth still increases rapidly with data dimensions, potentially impacting scalability.

**Questions:**

## Questions for the Authors
1.	In future real-hardware tests, is there a plan to optimize amplitude embedding to mitigate rapid circuit depth growth?
2.	Has there been any validation of FQN’s applicability to other time series tasks (e.g., forecasting or data imputation)? If so, would these tasks require architecture adjustments?

---

> ### Author Response · Authors · 2024-11-18
> **Response to Reviewer Zjmd**
>
> Questions:
> *1. In future real-hardware tests, is there a plan to optimize amplitude embedding to mitigate rapid circuit depth growth?*
>
> Response: If we get access to hardware in the future, then obviously we will optimize amplitude embedding to mitigate rapid circuit depth growth. However, for now, as we do not have access to quantum hardware capable of handling this level of implementation, we have not progressed in that particular direction for this study.
>
> *2. Has there been any validation of FQN’s applicability to other time series tasks (e.g., forecasting or data imputation)? If so, would these tasks require architecture adjustments?*
>
> Response: In this preliminary study, we have only worked with classification problems and have not tested forecasting or data imputation. We are currently working on making our model more robust, exploring further refinement in the design. For example, we are working on developing custom activation functions to scale the output of a Quanv1D layer beyond the range of -1 to +1. After we complete our current investigations, we will move on to other tasks—regression and imputation.
>
> For regression and imputation, we are predicting some form of change in the model. Depending on the task at hand, researchers usually change the head of the model for sequential data analysis, and we are expecting similar changes. However, we will try to keep the model fully quanvolutional for the parameter efficiency and self-regularizing effect.

---

### Official Review · Reviewer_ZJ9d · 2024-11-03

**Soundness:** 3
**Presentation:** 2
**Contribution:** 3
**Rating:** 6
**Confidence:** 3

**Summary:**

This paper proposes Fully Quanvolutional Networks (FQN) for time series classification, aiming to address limitations of previous quantum-classical hybrid models by developing an architecture composed entirely of quantum- inspired layers. The contributions of this paper include introducing the Quanv1D layer and utilizing Amplitude Embedding to reduce the number of required qubits for processing multi-dimensional data efficiently. Experimental results on 12 UEA time series datasets demonstrate that FQN achieves comparable or superior performance to existing models, including ModernTCN, with significantly fewer parameters.

**Strengths:**

- The manuscript effectively demonstrates the necessity of quantum operations and amplitude embedding, particularly through a self-regularization perspective, by analyzing gradient values.
- The manuscript thoroughly evaluates the proposed method against a comprehensive set of baseline models, including both quantum and classical approaches, and notably the current state-of-the-art, ModernTCN.

**Weaknesses:**

- The novelty of the proposed method appears somewhat incremental, as it primarily focuses on adapting amplitude embedding to the time series domain for improved scalability.
- While the authors provide a self-regularization perspective to explain the generalizability of the fully convolutional structure, additional analysis or insights into why this approach outperforms other quantum models are limited.

**Questions:**

- Could the authors provide insights into why FQN performs better than ModernTCN or QuanvNet?
- In some cases, FCN outperforms FQN across the 12 datasets. Beyond the similarity between the training and test sets, are there specific characteristic(s) of the datasets where FQN consistently excels?

---

> ### Author Response · Authors · 2024-11-18
> **Response to Reviewer ZJ9d (1)**
>
> Questions:
>
> *1. Could the authors provide insights into why FQN performs better than ModernTCN or QuanvNet?*
>
> Response: Due to recent works in the domain of ML, where increasing the number of parameters improves performance, we generally assume a model with more parameters to outperform a simpler model. However, if a model has enough parameters to ensure proper fit (assuming the model is fit for the task), the performance should not struggle when compared to a model with significantly higher parameters [1]. If we observe the cases where ModernTCN performed poorly compared to FQN, we will see that the model has clearly overfitted.
>
> | Dataset | FQN   |      |      | ModernTCN |       |       |
> |---------|-------|------|------|-----------|-------|-------|
> |         | Train | Val  | Test | Train     | Val   | Test  |
> | D1      | 0.27  | 0.29 | 0.31 | 0.00      | 0.01  | 0.20  |
> | D2      | 0.61  | 0.68 | 0.68 | 0.23      | 1.31  | 1.19  |
> | D3      | 0.55  | 0.59 | 0.57 | 0.18      | 0.52  | 0.51  |
> | D4      | 0.43  | 0.60 | 0.55 | 0.00      | 1.21  | 1.19  |
> | D5      | 0.40  | 0.44 | 0.43 | 0.00      | 0.00  | 0.06  |
> | D6      | 0.38  | 0.39 | 0.41 | 0.00      | 2.06  | 3.35  |
> | D7      | 0.48  | 0.82 | 0.57 | 0.00      | 12.73 | 3.67  |
> | D8      | 0.58  | 0.67 | 0.58 | 0.00      | 41.84 | 35.61 |
> | D9      | 0.36  | 0.37 | 0.37 | 0.00      | 2.63  | 1.48  |
> | D10     | 0.61  | 0.71 | 0.70 | 0.00      | 12.61 | 13.92 |
> | D11     | 0.55  | 0.75 | 0.69 | 0.01      | 2.03  | 2.27  |
> | D12     | 0.67  | 0.71 | 0.70 | 0.88      | 0.81  | 0.73  |
>
> This raises the question as to why ModernTCN performed well in D1-D5 and D9, where there is indication of overfitting. We believe, in these cases, the randomly split test set resembled the train set distribution closely which made overfitting advantageous. It is to be noted that parameter count for ModernTCN scales with the input and we have not deliberately increased the number of parameters. We have used the same embedding dimensions to maintain a fair comparison. All the other hyperparameters were taken according to the suggestions of the authors of ModernTCN [2].
>
> For QuanvNet, we believe the poor performance of the model is due to its construction. The first layer in QuanvNet is a trainable quantum layer while the rest of the network consists of a deep classical convolutional model [3]. In the recent literature on temporal convolutional networks (TCN), we have seen the importance of the very first layer in extracting key information from the original series, and the overall performance of the TCNs rely quite a bit on this layer. For example, the latest model in this domain - ModernTCN - uses a large kernel (k=51) to improve the effective receptive field of the embedding layer to ensure better extraction of temporal information. Unlike ModernTCN, the first layer of QuanvNet uses a constant kernel length of 3, which doesn’t allow the model to capture the temporal relationship in the data properly. On the other hand, our FQN model, built using our proposed quanvolution layer, permits variable kernel sizes, allowing the usage of a larger kernel to capture finer temporal variations during embedding. Additionally, the quantum layer of QuanvNet uses only one filter or circuit to generate the feature maps. Typically, CNN layers, and also our proposed Quanv1D layer, use multiple kernels to draw different information/patterns from the same data, helping the model to generalize better. Limiting the embedding layer (the first layer) to only one filter, as done in QuanvNet [3], can seriously limit the model performance, as seen from our experiments.
>
> [1] Aliferis, Constantin, and Gyorgy Simon. "Overfitting, Underfitting and General Model Overconfidence and Under-Performance Pitfalls and Best Practices in Machine Learning and AI." Artificial Intelligence and Machine Learning in Health Care and Medical Sciences: Best Practices and Pitfalls. Cham: Springer International Publishing, 2024. 477-524.
>
> [2] Luo, Donghao, and Xue Wang. "Moderntcn: A modern pure convolution structure for general time series analysis." The Twelfth International Conference on Learning Representations. 2024.
>
> [3] Rivera-Ruiz, Mayra Alejandra, et al. "1d quantum convolutional neural network for time series forecasting and classification." Mexican International Conference on Artificial Intelligence. Cham: Springer Nature Switzerland, 2023.

---

> ### Author Response · Authors · 2024-11-18
> **Response to Reviewer ZJ9d (2)**
>
> *2. In some cases, FCN outperforms FQN across the 12 datasets. Beyond the similarity between the training and test sets, are there specific characteristic(s) of the datasets where FQN constantly excels?*
>
> Response: Our goal for FQN was never to outperform FCN. We only wanted an equivalent convolutional layer in the quantum domain (Quanv1D), which will have similar capabilities as a classical convolutional layer (Conv1D). In order to test our proposed layer, we built a fully quanvolutional model (FQN) and compared it against a fully convolutional model (FCN). According to the performance data, FQN and FCN perform quite similarly. However, FQN offers a self-regularizing property as shown in Section 3.5. So, in datasets where the train set and test set are a bit too dissimilar or where generalization is hard, FQN tends to perform better, whereas FCN tends to overfit. For now, we have not noticed any other specific characteristic(s) of the datasets that allow FQN to consistently excel.
>
> As our study is quite preliminary, we have only tackled binary classification problems so far. We have found FQN to work consistently well on binary classification problems. Currently we are testing the performance of FQN on multiclass classification, and we should be able to present the results within a short time. At least for the datasets we have tested so far (selected randomly), FQN performs well on multiclass classifications as well.
>
> |                       | Test accuracy |         |           |              | Parameter ratio |        |           |              |
> |-----------------------|---------------|---------|-----------|--------------|-----------------|--------|-----------|--------------|
> |        Dataset        |      FQN      |   FCN   | ModernTCN |   QuanvNet   |       FQN       |   FCN  | ModernTCN |   QuanvNet   |
> |  EthanolConcentration |     29.14%    |  28.57% |   29.71%  |       -      |    x1 (4858)    |  x6.61 |  x107.34  |       -      |
> |      NerveDamage      |     99.51%    |  99.51% |   59.02%  |    98.54%    |    x1 (3440)    |  x3.42 |   x21.88  |    x27.02    |
> |  DiatomSizeReduction  |     98.13%    | 100.00% |   99.69%  |    89.06%    |    x1 (6098)    |  x6.37 |   x8.65   |    x15.32    |
> | HandMovementDirection |     34.47%    |  31.06% |   36.60%  |       -      |    x1 (1488)    |  x2.59 |   x96.82  |       -      |
> |    SyntheticControl   |     98.66%    |  98.83% |   96.83%  |    96.00%    |    x1 (2988)    | x13.02 |   x11.12  |    x31.62    |
> |        Average        |     71.98%    |  71.60% |   64.37%  | _Incomplete_ |        x1       |  x6.40 |   x49.16  | _Incomplete_ |
>
> Here, the values of QuanvNet are incomplete as it can only handle univariate time series.

---

> ### Author Response · Authors · 2024-11-18
> **Response to Reviewer ZJ9d (3)**
>
> We would also like to address some weaknesses, as we believe we can provide some clarifications regarding them.
>
> Weaknesses:
> *1. The novelty of the proposed method appears somewhat incremental, as it primarily focuses on adapting amplitude embedding to the time series domain for improved scalability.*
>
> Response: While amplitude embedding is a key part of our algorithm, we do not consider it a contribution. Our main contribution was designing a trainable and modular (can be used anywhere in the model depending on the input shape) quanvolution layer capable of handling any arbitrary dimensional data. The existing 1D quanvolution layer in the literature uses only one circuit/filter to generate a fixed number of feature maps. We worked on this immutability and ensured that our proposed Quanv1D layer is just as dynamic as Conv1D and can be used freely by changing hyperparameters similar to Conv1D. As our work is theoretical in nature, we could not properly show the widely sought-after “quantum advantage” without the required hardware. However, we tried to simulate practical scenarios to show some areas where Quanv1D shines. For example, we simulated statistical noise (finite shot-based learning) and showed that Quanv1D is still effective in Section 3.4. Also, our proposed algorithm has inherent self-regularization, which helped it generalize on many datasets. Finally, due to our design choices inside the layer, we were able to make Quanv1D very parameter efficient. There is a linear relation between the number of qubits in a circuit and feature maps. For example, a higher number of input channels typically requires more qubits (and hence, more parameters), but Quanv1D minimizes this by utilizing all available qubits in its circuits, thereby reducing the total number of filters needed.
>
> Our secondary contribution comes from the model architecture itself. In the literature, researchers only use one quanvolution layer at the very top, and the rest of the network only consists of different classical layers. Thus, these hybrid networks are heavily reliant on classical layers to extract meaningful information from the quanvolution layer. To truly understand the impact of quanvolution in data learning tasks, we built an architecture that only contains quantum layers. We tested the applicability of this network against its classical counterpart, FCN, and found that FQN was roughly on par with FCN in terms of accuracy. Additionally, the self-regularization property of FQN combined with its parameter efficiency makes FQN an attractive choice when compared to FCN and even the time-series SOTA, ModernTCN. One distinct advantage FQN has over FCN is that the depth of the FQN network can be increased without a significant increase in parameters due to our use of increamental dilation in the propagation layers. This increases the network’s receptive field as the depth grows, which is helpful for recognizing longer sequential patterns.
>
> We acknowledge that our writing was not up to par, which is why the incorporation of amplitude embedding overshadowed some of our contributions. We are currently working on the writing.

---

> ### Author Response · Authors · 2024-11-18
> **Response to Reviewer ZJ9d (4)**
>
> *2. While the authors provide a self-regularization perspective to explain the generalizability of the fully convolutional structure, additional analysis or insights into why this approach outperforms other quantum models are limited.*
>
> Response: In the literature, there are two variants of quantum convolution: circuit-based and kernel-based (also referred to as quanvolution). Circuit-based quantum convolution involves converting a full CNN model into a quantum circuit. In such circuits, different quantum operations, according to the authors, mimic convolution-related layers (like Conv2D and pooling) [1]. However, this is different from a typical convolution operation that applies to individual input patches. On the other hand, kernel-based quantum convolution, or quanvolution, operates like a convolutional layer and can be implemented through quantum circuits and their unitary operations [2]. Simply put, in kernel-based quantum convolution, quantum circuits act as kernels and replace the typical convolutional filters in a convolutional layer. Our work pertains to the second variant, and as such, models related to the first variant have not been considered in this study.
>
> Currently, the number of studies focusing on quanvolution in time series analysis is very limited. Most works [3–7] used the image-based variant of quanvolution, transforming the time series data into images first (like using a scalogram or a spectrogram). With the help of a systematic search through four databases (Scopus, Web of Science, IEEE Xplore, and ACM Digital Library), we found only one study that directly used a 1D quanvolution-based model for time series analysis [8]. We included this model, QuanvNet, in the comparison, and FQN overall outperformed it. One major limitation that QuanvNet has is that it can only work with one-channel data, i.e., univariate data. As such, we could not compare QuanvNet in multivariate scenarios.
>
> Some insights on why FQN outperforms existing quantum models are discussed in the response of question 1.
>
> [1] Cong, Iris, Soonwon Choi, and Mikhail D. Lukin. "Quantum convolutional neural networks." Nature Physics 15.12 (2019): 1273-1278.
>
> [2] Henderson, Maxwell, et al. "Quanvolutional neural networks: powering image recognition with quantum circuits." Quantum Machine Intelligence 2.1 (2020): 2.
>
> [3] Li, Yue, et al. "Detection and Identification of Power Quality Disturbance Signals in New Power System Based on Quantum Classic Hybrid Convolutional Neural Networks." International Conference on Data Security and Privacy Protection. Singapore: Springer Nature Singapore, 2024.
>
> [4] Savla, Aansh, et al. "GQNN: Greedy Quanvolutional Neural Network Model." International Conference on Image Processing and Capsule Networks. Cham: Springer International Publishing, 2022.
>
> [5] Sridevi, S., et al. "Quanvolution neural network to recognize arrhythmia from 2D scaleogram features of ECG signals." 2022 International Conference on Innovative Trends in Information Technology (ICITIIT). IEEE, 2022.
>
> [6] Prabhu, Sharanya, et al. "QuCardio: Application of Quantum Machine Learning for Detection of Cardiovascular Diseases." IEEE Access 11 (2023): 136122-136135.
>
> [7] Yang, Chao-Han Huck, et al. "Decentralizing feature extraction with quantum convolutional neural network for automatic speech recognition." ICASSP 2021-2021 IEEE International Conference on Acoustics, Speech and Signal Processing (ICASSP). IEEE, 2021.
>
> [8] Rivera-Ruiz, Mayra Alejandra, et al. "1d quantum convolutional neural network for time series forecasting and classification." Mexican International Conference on Artificial Intelligence. Cham: Springer Nature Switzerland, 2023.

---

> > ### Comment · Reviewer_ZJ9d · 2024-12-02
> >
> > I appreciate the detailed and comprehensive rebuttal from the authors.
> > The additional context has strengthened the understanding of their contributions.
> > I will take these points into careful consideration during my final evaluation.

---

### Comment · Area_Chair_5nYt · 2024-11-25

Dear reviewers,

A reminder that **November, 26** is the last day to interact with the authors, before the private discussion with the area chairs. At the very least, please acknowledge having read the rebuttal (if present). If the rebuttal was satisfying, please improve your score accordingly. Finally, if you have concerns that might be solved in time, this is the last chance before moving on to the next phase.

Thanks,
The AC

---

### Author Response · Authors · 2024-11-28
**Revision Summary**

We would like to thank the reviewers for their insightful comments. The feedback has helped us significantly improve our manuscript. We have uploaded an updated version of the manuscript and would like to summarize the changes in this comment.

Changes to the manuscript -

1. We have updated the *Abstract*, making it less focused on amplitude embedding and instead focusing on our actual contributions.

2. We have rewritten the *Introduction* section to provide greater clarity on the research gap and our contribution to the domain. We have segmented the section into *Background*, *Motivation*, and *Contribution* subsections. The *Background* subsection presents an overview of the current research trends in the domain of quantum convolution. The *Motivational* subsection expands upon the background section, highlighting the limitations of current methods and the areas we aim to fill. Finally, in the *Contribution* subsection, we have talked about the novelties of our work from a non-technical perspective. We have pushed all technical information to the *Method* section.

3. We have updated the writing in the *Method* section to enhance clarity. Together, the *Introduction* and the *Method* aid in differentiating our proposed quanvolution algorithm from previous ones. There is one technical update in this section. We considered $\phi$ as a trainable parameter in the previous version of the manuscript due to its phase-shifting ability but found that gradient updates did not affect it. In this updated version, we kept $\phi$ untrainable. Removing $\phi$ from the parameter list changed (decreased) the parameter count for FQN in all the subsequent sections. However, theoretically, this change should not impact performance, and our reruns confirm this, as the accuracy remains unchanged.

4. In the previous manuscript, we presented results from experiments conducted on 12 datasets, all of which belonged to binary classification tasks. In the updated manuscript, we have added 8 more datasets for multiclass classification problems. Same as before, we selected these datasets from the UEA and UCR archives randomly but prioritized domain diversity.

5. In the revised manuscript, we have included an analysis of FCN's performance when trained with the same parameters as FQN, as recommended by reviewer ya86. Sections 3.3 and 3.4 detail this analysis, which confirms that our chosen ansatz, and not fewer parameters, is the source of FQN's self-regularization.

6. We have revamped Section 3.5 to better formulate the problem and explain the findings. In the previous version of the manuscript, we couldn’t run the finite shot model up to theoretical accuracy for the PowerCons dataset due to computational constraints (both VRAM and time). We have now used a different GPU and were able to run it up to 1000 shots, where PowerCons reached within 1% of the analytical model’s accuracy (previously the gap was around 10%). Furthermore, we implemented finite shot analysis for two additional datasets. We selected all the datasets for this section at random.

7. We have updated the *Concluding Remarks* section to include the implications of our study in the broader domain. We have adjusted the *Limitations* and *Future Work* subsections to reflect the changes in the updated manuscript.

8. We added a *Preliminaries* section in the *Appendix* as suggested by reviewer kez7 to aid researchers unfamiliar with the field of quantum computing in navigating the paper better.

9. We have slightly updated the overall *Appendix* section to support the changes made in the main manuscript.

---

### Meta-Review · Area_Chair_5nYt · 2024-12-16

**Metareview:**

The paper introduces a novel "quanvolutional" layer, the equivalent of a 1D convolutional filter implemented via quantum operations, and a novel fully quantum architecture built on this novel layer.

The reviewers are almost unanimously negative on the paper, highlighting several concerns. In general, I would say the paper is tackling an important topic, but the paper failed to properly position itself in terms of related works, and the experimental evaluation and writing were considered insufficient. Some of these points have also been validated by the authors themselves.

I agree with the reviewers that the paper requires more work to be presentable, and as a result, I suggest to reject it for this conference.

**Additional Comments On Reviewer Discussion:**

- **Reviewer kez7** was concerned about the choice of datasets and baselines, as well as some unclear motivation. Rebuttal showcased some additional experiments, but it was considered insufficient overall.

- **Reviewer fVyg** had similar concerns, and they also highlighted concerns in the related works and in the lack of consideration of real hardware. I consider the last point minor (due to the current state of quantum neural networks), so it weighted less in my evaluation.

- **Reviewer YA86** was concerned about writing, related works, and the experimental setup. Also in this case, the rebuttal was considered insufficient. In particular, the authors themselves agree that writing is not up to par for a submission. I took this point into serious consideration for the final evaluation.

- **Reviewer Zjmd** had similar concerns but they did not interact during the rebuttal. The overall review seemed hastily written and I ignored it for the final decision, as it was mostly overlapping with the other reviewers.

- **Reviewer ZJ9d** was concerned about lack of novelty and some missing clarifications in the paper. They answered briefly in the rebuttal but there was no real discussion. I personally find the answer from the authors convincing, and this was the main positive point I considered in the final evaluation.

---

### Decision · Program_Chairs · 2025-01-22

Reject